# IndicSuperTokenizer: An Optimized Tokenizer for Indic Multilingual LLMs

## Abstract

Tokenizers play a crucial role in determining the performance, training efficiency, and the inference cost of Large Language Models (LLMs). Designing effective tokenizers for multilingual LLMs is particularly challenging due to diverse scripts and rich morphological variation. While subword methods such as Byte Pair Encoding (BPE) are widely adopted, their effectiveness in multilingual settings remains underexplored. We present IndicSuperTokenizer, a tokenizer for Indic multilingual LLMs, that combines both subword and multi-word tokenization, along with language-specific pre-tokenization, leading to more linguistically aligned tokens and achieving a new state-of-the-art in fertility score. Evaluated across English, 22 Indian languages and code data, our tokenizer improves the average fertility score by $39.5\%$ over LLaMA4 and by $18\%$ over Sutra (the current best). This translates to $44\%$ improvement in inference throughput over LLaMA4 while maintaining comparable performance on English and Indic benchmarks. We also present detailed ablations across tokenizer training data size, vocabulary size, merging techniques, and pre-tokenization strategies, demonstrating the robustness of our design choices.

## 1 Introduction

Large Language Models (LLMs) (Touvron et al., 2023; Grattafiori et al., 2024; Abdin et al., 2025; Guo et al., 2025; Yang et al., 2025; Team et al., 2025) rely on the crucial step of tokenization, the process of converting raw text into discrete units called *tokens*. A key metric for evaluating tokenizers is the "fertility score" (or token-to-word ratio) (Ali et al., 2024) where, a lower fertility score is desirable due to more efficient (and hence cheaper) LLM training and inference. Among the many proposed approaches, subword tokenization schemes such as BPE (Sennrich et al., 2016a), Unigram (Kudo, 2018), WordPiece (Song et al., 2021), and their byte-level extensions have become the de facto choice. However, tokenization remains an understudied topic within the LLM literature (Dagan et al., 2024; Mielke et al., 2021), especially in multilingual settings (Petrov et al., 2023), where, skewed fertility scores across languages, often lead to concerns around fairness, high inference latency, cost and context size. With 22 constitutionally recognized languages[1], these issues are especially pronounced for Indic languages comprising multiple scripts and a rich morphology. Our analysis suggests that tokenizers of popular multilingual tokenizers, largely designed for English, could produce fertility scores as high as 10.5 (LLaMA-4 tokenizer for Oriya; Table 3) for Indic languages, far worse than the near-ideal scores achieved for English. This leads to longer token sequences, higher compute overheads, and poor alignment with linguistic units like morphemes and compounds.

Designing an efficient tokenizer involves making careful choices around the size of the vocabulary (of tokens), tokenizer training data, the tokenization approach, and, doing this across languages is nontrivial. Our work concerns the broader problem of training an effective multilingual tokenizer where we address five core research questions: *i).* How can we improve low-resource language performance without degrading high-resource performance? *ii).* Should we train language-specific tokenizers and merge them, or adopt a unified joint training paradigm? *iii).* How do we determine an appropriate multilingual training data distribution? *iv).* What is the role of pre-tokenization in

---

[1] https://en.wikipedia.org/wiki/Languages_with_official_recognition_in_India

| Language | Tokenizer | Tokens |
|----------|-----------|--------|
| English | IST | I wake up early in the morning and get ready for school. My mother makes tea and puts |
|          | Sutra | I wake up early in the morning and get ready for school. My mother makes tea and puts |
|          | Gemma | I wake up early in the morning and get ready for school. My mother makes tea and puts |
| Hindi | IST | मैं सुबह जल्दी उठ जाता हूँ और तैयार हो जाता हूँ। माँ चाय बनाती हैं और नाश्ता लगा देती हैं। नाश्ता करने के बाद |
|       | Sutra | मैं सुबह जल्दी उठ जाता हूँ और तैयार हो जाता हूँ। माँ चाय बनाती हैं और नाश्ता लगा देती हैं। नाश्ता करने के बाद |
|       | Gemma | मैं सुबह जल्दी उठ जाता हूँ और तैयार हो जाता हूँ। माँ चाय बनाती हैं और नाश्ता लगा देती हैं। नाश्ता करने के बाद मैं |
| Bengali | IST | এই গ্রামে শিক্ষার জন্য কাজ করা হয়, এবং সেখানে সাক্ষরতার হার কম বয়সী শিশুদের মধ্যে দেখা যায়। স্কুল |
|         | Sutra | এই গ্রামে শিক্ষার জন্য কাজ করা হয়, এবং সেখানে সাক্ষরতার হার কম বয়সী শিশুদের মধ্যে দেখা যায়। |
|         | Gemma | এই গ্রামে শিক্ষার জন্য কাজ করা হয়, এবং সেখানে সাক্ষরতার হার কম বয়সী শিশুদের মধ্যে দেখা যায়। স্কুল |
| Tamil | IST | இந்த கிராமத்தில் ஒரு பள்ளி உள்ளது என்று கூற மக்கள் சொல்லுகிறார்கள். அந்த பள்ளியில் |
|       | Sutra | இந்த கிராமத்தில் ஒரு பள்ளி உள்ளது என்று கூற மக்கள் சொல்லுகிறார்கள். அந்த பள்ள |
|       | Gemma | இந்த கிராமத்தில் ஒரு பள்ளி உள்ளது என்று கூற மக்கள் சொல்லுகிறார்கள். அந்த பள்ளியில் |

Figure 1: IndicSuperTokenizer (IST) captures superwords (e.g. "wake up", "in the morning") and avoids fragmenting Indic words (see for e.g. Bengali, Tamil).

multilingual tokenizer training? *v).* Do multi-word expressions provide measurable benefits when incorporated into the tokenizer vocabulary and what is the effective way to learn these multi-words? Through our controlled experiments and ablations, we provide a systematic recipe for training equitable and culturally inclusive multilingual tokenizers.

In this work, we present IndicSuperTokenizer, an efficient tokenizer for Indic LLMs, that achieves state-of-the-art fertility scores across 22 Indic languages, English, and code. Our design choices are grounded in detailed ablations and our tokenizer combines linguistically grounded pre-tokenization with a two-stage subword–superword learning process (Liu et al., 2025b), yielding a more compact and semantically faithful vocabulary. Figure 1 illustrates some examples where our approach avoids fragmenting common words or idiomatic phrases into unnatural subunits across different languages. We make the following contributions:

- We present IndicSuperTokenizer, a state-of-the-art tokenizer for Indic LLMs, systematically benchmarking it against popular multilingual baselines.
- We study the impact of vocabulary size, training data, and language-specific pre-tokenization choices on fertility score, showing that careful pre-tokenization outweighs naive vocabulary scaling.
- To the best of our knowledge, we are the first to carry out a comprehensive benchmarking of a tokenizer across multiple intrinsic quality measures, as well as to study its downstream impact on task performance and LLM inference efficiency in both pretraining from scratch as well as continual pretraining settings.

## 2 RELATED WORK

**Tokenization Algorithms.** Tokenization strategies differ in both theory and practice. While alternate sub-word tokenization algorithms have been explored in the past such as WordPiece (Song et al., 2021), Unigram LM (Kudo & Richardson, 2018), Byte Pair Encoding (BPE) remains the most widely adopted. Originally developed for compression (Gage, 1994) and later adapted for neural MT (Sennrich et al., 2016b), BPE merges frequent character pairs to balance coverage with efficiency. Its variants aim to address inefficiencies: PickyBPE (Chizhov et al., 2024) discards uninformative merges to improve vocabulary utility, while Scaffold-BPE (Lian et al., 2024) iteratively prunes low-frequency scaffold tokens to reduce imbalance and enhance downstream performance. Recent extensions like SuperBPE (Liu et al., 2025a) expand beyond word boundaries, jointly learning subwords and multi-word "superwords" yielding improved compression and inference efficiency in a 2-stage curriculum. BoundlessBPE (Schmidt et al., 2024), another contemporary work, relaxes

the Pre-tokenization word boundary constraint in an single stage learning step. Our work compares these two recent approaches and show that two-stage curriculum preserves subword coverage while capturing larger semantic units in morphologically rich Indian languages.

**Multilingual Tokenizers.** Multilingual tokenization faces challenges from script diversity, morphology, and structural variation. Comparative studies show that vocabulary size and construction strategies strongly affect performance for morphologically rich languages (Karthika et al., 2025a), while inefficiencies in underrepresented ones, such as Ukrainian, translate to higher fertility and computational costs (Maksymenko & Turuta, 2025). Tokenization also influences how multilingual models encode morphology, as demonstrated in mT5 vs. ByT5 (Dang et al., 2024). For Indic languages, tailored resources (Kakwani et al., 2020) and IndicBERT (AI4Bharat, 2022) highlight the value of domain-specific tokenization. Recent benchmarks further reveal economic implications, with BLOOM's tokenizer achieving the best cost efficiency among popular multilingual LLMs (ADA Sci, 2024). Together, these studies show that current multilingual tokenizers fragment low-resource and morphologically rich languages, motivating approaches like ours that combine normalization, language-tailored pre-tokenization, and multi-word learning to achieve better efficiency and fairness in Indic languages. Tokenization for Indic languages presents unique challenges due to their linguistic diversity, rich morphology, and script multiplicity.

**Pre-tokenization.** Pre-tokenization plays a pivotal role in shaping token boundaries, directly influencing both compression efficiency and reasoning performance (Xue et al., 2024). Sentence-Piece (Kudo & Richardson, 2018) introduced a language-agnostic approach by treating input as raw streams, effective for languages without whitespace boundaries. More recent approaches like BoundlessBPE (Schmidt et al., 2024) relax pre-token constraints to improve frequency distributions, while regex-based designs continue to prove crucial for capturing script-specific structures.

## 3 INDICSUPERTOKENIZER (IST)

Language modeling involves estimating the probability distribution over text sequences, $P(S)$, where $S$ may represent a sentence, paragraph, or document. To achieve this, the text is first converted into a sequence of discrete tokens through a tokenization function $g(S) = X = (X_1, X_2, \ldots, X_n) \in V^n$, where $V$ denotes the vocabulary and $n$ the sequence length. Tokenizers can be open-vocabulary, ensuring any string can be represented (e.g., byte-level), or closed-vocabulary, where unseen text maps to an out-of-vocabulary symbol (e.g., word lists) (Rae et al., 2021). In our work, we adopt an open-vocabulary approach that combines byte-pair encoding (BPE) with a UTF-8 byte fallback, following Radford et al. (2018). In this section, we describe our tokenizer training and evaluation approach.

### 3.1 TOKENIZER TRAINING

With the aim of improving fertility in Indic languages and scripts, we follow the curriculum principles as in Liu et al. (2025a). Specifically, we have:

*Stage 1 (Subword Learning):* Training begins with standard byte-pair encoding (BPE) applied after whitespace pre-tokenization. This ensures that merges occur only within word boundaries, allowing the tokenizer to learn fine-grained *subword units* such as roots, affixes, and common morphemes. Stage 1 continues until the vocabulary reaches a pre-defined *transition point* $t$ ($< |V|$).

*Stage 2 (Superword Learning):* After reaching $t$, training resumes without whitespace constraints, allowing BPE to merge across word boundaries. This enables the formation of *superwords*, frequent multiword expressions or collocations (e.g., "one of the", "number of"), improving compression and reducing token counts for common phrases.

This two-stage tokenizer training is particularly effective for morphologically rich languages and scripts with complex variations where, meaningful subwords are first anchored and then composed into frequent multiword units.

## 3.2 PRE-TOKENIZATION

Pre-tokenization segments raw text before subword learning to improve token consistency and efficiency. We combine regex-based, Unicode normalization, and morphology-aware strategies. Unicode-aware regex separates punctuation, handles numeric groups, and aligns tokens with semantic units. NFKC normalization standardizes visually identical characters, reducing sparsity (Table 16 illustrates the effect of normalization). Morphology-aware segmentation decomposes words into roots and affixes to capture recurring morphemes. While we experimented with morphology-aware segmentation, including them in tokenization without impacting the latency is non-trivial (Refer to Appendix C.2 for details). In contrast to SuperBPE, in our Stage 1 pre-tokenization step we replace GPT-2 rules with LLaMA-4 regex for script-agnostic segmentation, improving token-to-word ratios by 38–40% (See Table 1) on Indic scripts. Stage 2 relaxes whitespace constraints to form multiword tokens capturing collocations and idioms. This design produces a script-robust tokenizer that efficiently supports multiword learning across English and Indic languages. However, unconstrained merging risks producing tokens that cross sentence boundaries,which destabilizes generation and distorts end-of-sentence probabilities. To mitigate this, we introduce sentence-level boundary constraints: merges are free within sentences but are disallowed across sentence delimiters.

Table 1: Fertility scores showing LLaMA-4 regex outperforms GPT-2 in stage-1 tokenizer training.

| Regex | as | bn | brx | code | doi | eng | gom | gu | hi | kas | kn | mai | ml | mni | mr | nep | or | pa | san | sat | snd | ta | te | urd |
|---|---|---|---|---|---|---|---|---|---|---|---|---|---|---|---|---|---|---|---|---|---|---|---|---|
| GPT-2 | 4.36 | 4.72 | 4.67 | 1.57 | 2.88 | 1.32 | 3.95 | 4.12 | 3.47 | 2.47 | 5.95 | 3.17 | 7.08 | 3.30 | 4.86 | 4.37 | 4.44 | 3.28 | 5.97 | 2.71 | 1.30 | 6.53 | 5.61 | 1.29 |
| LLaMA-4 | 1.83 | 1.74 | 1.99 | 1.54 | 1.56 | 1.33 | 2.17 | 1.83 | 1.36 | 1.36 | 2.15 | 1.56 | 2.24 | 2.27 | 1.61 | 1.59 | 1.65 | 1.47 | 2.51 | 3.60 | 1.45 | 2.07 | 1.83 | 1.47 |

## 3.3 TRAINING DATA AND VOCABULARY

Training a multilingual tokenizer involves careful design choices on the vocabulary size, language (or language script)-wise vocabulary distribution, and training data mix. We evaluate different vocabulary allocation strategies (Section 4.4) and conduct detailed ablations (Section 5) to inform these design choices. The final IndicSuperTokenizer that we train uses a shared vocabulary of 200K tokens, distributed across language scripts (Figure 2), and is trained on 10GB of multilingual high quality data curated from OLMo (OLMo et al., 2025), Wikipedia[2], books, PDFs, Common Crawl, and the Sangraha dataset (Khan et al., 2024).

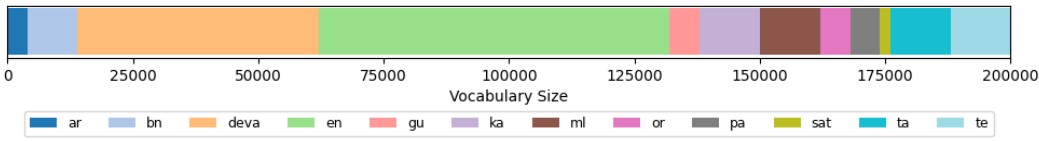

Figure 2: Vocabulary size distribution across language scripts. See Appendix A.1 for script details.

## 3.4 BASELINES

We benchmark against 9 tokenizers, comprising: *i) Indic-focused tokenizers:* tokenizers designed primarily for Indian languages: Sutra (Tamang & Bora, 2024) and Sarvam-2B (Team, 2024b) (referred as Sarvam). *ii) Good Indic support tokenizers:* multilingual tokenizers with demonstrated capabilities for Indic languages: Gemma-3-27B-it (Team et al., 2025) (referred as Gemma-3), GPT-oss (OpenAI, 2025) and LLaMA-4 (AI, 2025b). *iii) General tokenizers:* tokenizers of widely-used general-purpose LLMs: Qwen3-32B (Team, 2024a) (referred as Qwen-3), LLaMA-3.2-1B (Dubey et al., 2024), Mistral-Nemo (AI, 2024) and DeepSeek-R1 (AI, 2025a).

## 3.5 METRICS

We employ four intrinsic metrics capturing different aspects of token efficiency and informativeness: (i) Fertility score (Rust et al., 2021; Scao et al., 2022), measuring vocabulary granularity; (ii) Normalized Sequence Length (NSL) (Dagan et al., 2024), quantifying sequence compression relative

---

[2]https://en.wikipedia.org/wiki/

to a base tokenizer; (iii) Rényi's entropy and efficiency (Zouhar et al., 2023), assessing information density; and (iv) Bytes per token (Kocetkov et al., 2022), reflecting memory and storage efficiency. We report micro-average per line at the language level. More details on the metrics and definitions in Section D in Appendix.

## 3.6 EVALUATION FRAMEWORK

We construct an evaluation set spanning 22 Indic languages, English, and code, curated from the same sources as the training corpus. Table 2 reports dataset statistics: text volume, number of lines, and average words per line per language. All metrics are computed at the line level and aggregated to the language level.

We also develop a modular evaluation framework supporting HuggingFace[3], SentencePiece[4], and TikToken[5] tokenizers along with a comprehensive set of intrinsic metrics, including, Fertility score, normalized sequence length (NSL), Rényi entropy and efficiency, and bytes per token. We will release both the evaluation dataset and the framework for reproducible benchmarking and fair comparison of multilingual tokenizers.

Table 2: Evaluation corpus statistics across 22 Indic languages, English, and code. We report standard ISO codes here. See Section A.1 for the actual language name.

| | as | bn | brx | code | doi | eng | gom | gu | hi | kas | kn | mai | ml | mni | mr | nep | or | pa | san | sat | snd | ta | te | urd |
|---|---|---|---|---|---|---|---|---|---|---|---|---|---|---|---|---|---|---|---|---|---|---|---|---|
| Size (MB) | 56 | 562 | 13 | 4 | 1 | 148 | 67 | 83 | 422 | 30 | 252 | 60 | 311 | 34 | 152 | 82 | 46 | 144 | 110 | 0.47 | 38 | 502 | 486 | 38 |
| # Lines (K) | 65 | 681 | 19 | 118 | 2 | 449 | 135 | 91 | 545 | 21 | 273 | 129 | 337 | 64 | 257 | 126 | 59 | 198 | 139 | 1 | 69 | 658 | 773 | 27 |
| Avg W/Line | 51 | 47 | 41 | 3 | 43 | 56 | 37 | 59 | 59 | 145 | 41 | 39 | 35 | 43 | 33 | 40 | 45 | 57 | 36 | 26 | 70 | 32 | 32 | 185 |

# 4 EXPERIMENTS AND RESULTS

## 4.1 INTRINSIC EVALUATION OF TOKENIZERS

We achieve SOTA performance across 9 tokenizers for fertility score in consideration (see Table 3 for Indic focused or good Indic support tokenizers and an extended version Table 24 in Appendix for the rest). As shown in Table 3, IndicSuperTokenizer consistently achieves the lowest ratios across all evaluated languages, which reflects the degree of fragmentation. Bytes-per-token (Appendix D.2) measures the average raw text bytes per token, indicating information density and sequence compactness. Table 7 shows that IndicSuperTokenizer achieves consistently higher values across languages. See Appendix D.2 for details.

Table 3: Fertility score (↓) comparison for Indic focused and Good support tokenizers across languages here. IST performs best in 20 of 24 languages. An extended version in Table 24 (Appendix).

| Tokenizer (↓) | as | bn | brx | code | doi | eng | gom | gu | hi | kas | kn | mai | ml | mni | mr | nep | or | pa | san | sat | snd | ta | te | urd |
|---|---|---|---|---|---|---|---|---|---|---|---|---|---|---|---|---|---|---|---|---|---|---|---|---|
| Gemma-3 | 2.65 | 1.69 | 2.84 | 1.79 | 1.69 | 1.39 | 2.60 | 2.50 | 1.47 | 1.48 | 3.34 | 1.91 | 3.45 | 2.07 | 2.03 | 2.03 | 4.42 | 2.83 | 3.37 | 5.16 | 2.03 | 2.50 | 2.94 | 1.44 |
| GPT-OSS | 2.66 | 2.41 | 3.17 | 1.51 | 1.89 | 1.33 | 2.73 | 2.37 | 1.72 | 1.58 | 3.34 | 2.01 | 3.51 | 2.41 | 2.61 | 2.10 | 6.26 | 2.71 | 3.89 | 13.01 | 1.76 | 3.18 | 3.13 | 1.51 |
| LLaMA-4 | 4.40 | 2.93 | 3.34 | 1.46 | 2.00 | 1.34 | 2.84 | 3.37 | 1.83 | 1.72 | 4.23 | 2.28 | 4.95 | 2.73 | 2.79 | 2.46 | 10.51 | 3.23 | 4.12 | 9.04 | 2.13 | 5.87 | 4.53 | 1.76 |
| Sarvam | 4.24 | 1.91 | 2.92 | 2.14 | 1.85 | 1.66 | 3.01 | 2.11 | 1.53 | 1.91 | 2.53 | 2.11 | 3.19 | 4.60 | 1.94 | 2.35 | 2.43 | 1.67 | 3.78 | 13.07 | 7.62 | 2.49 | 2.63 | 7.93 |
| Sutra | 2.12 | 2.07 | 3.06 | 2.12 | 1.78 | 1.17 | 2.68 | 2.15 | 1.62 | 1.48 | 2.71 | 2.08 | 3.10 | 2.40 | 2.18 | 2.01 | 2.24 | 1.50 | 3.76 | 2.03 | 2.23 | 2.58 | 2.77 | 1.55 |
| IST | **1.85** | 1.74 | **2.04** | 1.47 | **1.45** | **1.12** | **2.17** | **1.77** | **1.23** | **1.21** | **2.19** | **1.58** | **2.30** | 2.28 | **1.63** | **1.62** | **1.65** | **1.39** | **2.59** | **3.72** | **1.45** | **2.12** | **1.88** | 1.44 |

Table 4: NSL score (↓) comparison for Indic focused and Good support tokenizers across languages here. IST performs best in 23 of 24 languages. An extended version in Table 23 (Appendix).

| Tokenizer (↓) | as | bn | brx | code | doi | eng | gom | gu | hi | kas | kn | mai | ml | mni | mr | nep | or | pa | san | sat | snd | ta | te | urd |
|---|---|---|---|---|---|---|---|---|---|---|---|---|---|---|---|---|---|---|---|---|---|---|---|---|
| Gemma-3 | 0.63 | 0.59 | 0.87 | 1.31 | 0.91 | 1.06 | 0.94 | 0.76 | 0.83 | 0.93 | 0.81 | 0.89 | 0.73 | 0.81 | 0.76 | 0.83 | 0.44 | 0.89 | 0.84 | 0.59 | 0.99 | 0.45 | 0.67 | 0.85 |
| GPT-oss | 0.63 | 0.83 | 0.95 | 1.03 | 0.96 | 1.00 | 0.96 | 0.71 | 0.94 | 0.95 | 0.79 | 0.90 | 0.72 | 0.89 | 0.94 | 0.85 | 0.60 | 0.85 | 0.94 | 1.43 | 0.83 | 0.56 | 0.71 | 0.88 |
| Sutra | 0.55 | 0.74 | 0.93 | 2.09 | 0.92 | 0.89 | 0.96 | 0.68 | 0.92 | 0.91 | 0.67 | 0.94 | 0.65 | 0.92 | 0.84 | 0.82 | 0.24 | 0.51 | 0.91 | 0.26 | 1.10 | 0.47 | 0.59 | 0.90 |
| Sarvam | 0.99 | 0.66 | 0.91 | 1.50 | 1.00 | 1.27 | 1.13 | 0.64 | 0.85 | 1.19 | 0.62 | 0.99 | 0.65 | 2.19 | 0.72 | 0.96 | 0.24 | 0.54 | 0.93 | 1.45 | 3.63 | 0.45 | 0.56 | 4.25 |
| IST | **0.45** | **0.60** | **0.65** | **0.94** | **0.78** | **0.85** | **0.82** | **0.54** | **0.68** | **0.80** | **0.53** | **0.76** | **0.50** | **0.91** | **0.61** | **0.67** | **0.18** | **0.45** | **0.66** | **0.45** | **0.72** | **0.38** | **0.44** | **0.86** |

---

[3] https://github.com/huggingface/tokenizers
[4] https://github.com/google/sentencepiece
[5] https://github.com/openai/tiktoken

Normalized sequence length (Appendix D.3) quantifies tokenized sequence length relative to a base tokenizer, indicating relative compression efficiency. Table 4 shows that IndicSuperTokenizer achieves shorter normalized sequences across languages. Rényi's entropy quantifies the uncertainty of token distributions, while Rényi's efficiency normalizes entropy by vocabulary size to assess utilization. Table 6 shows that IndicSuperTokenizer achieves superior efficiency across languages, reflecting effective and balanced token allocation.

Table 5: Inference latency comparison of 1B models trained with LLaMA-4 and IST tokenizers.

| Model | TTFT (ms) ↓ | OTPT (tokens/s) ↑ |
|---|---|---|
| LLaMA-4 | 19.17 ± 0.15 | 117.99 |
| IST | **18.98 ± 0.36** | **169.42** |

Table 6: Rényi's Entropy and Efficiency across top Indic tokenizers. Higher efficiency indicates better balance between vocabulary capacity and token usage.

| | Gemma-3 | GPT-oss | LLaMA-4 | Sarvam | Sutra | IST |
|---|---|---|---|---|---|---|
| Entropy ↓ | 20.70 | 20.81 | 21.09 | 20.71 | 20.62 | **20.42** |
| Efficiency ↑ | 0.22 | 0.19 | 0.14 | 0.21 | 0.23 | **0.28** |

Table 7: Bytes-per-token score (↑) comparison for Indic focused and Good support tokenizers across languages here. IST performs best in 22 of 24 languages.

| Tokenizer (↑) | as | bn | brx | code | doi | eng | gom | gu | hi | kas | kn | mai | ml | mni | mr | nep | or | pa | san | sat | snd | ta | te | urd |
|---|---|---|---|---|---|---|---|---|---|---|---|---|---|---|---|---|---|---|---|---|---|---|---|---|
| Gemma-3 | 6.37 | 10.45 | 5.87 | 2.33 | 6.75 | 4.36 | 5.29 | 6.31 | 9.16 | 7.01 | 6.73 | 6.42 | 7.57 | 6.23 | 8.90 | 8.31 | 3.76 | 4.62 | 6.66 | 2.59 | 3.87 | 9.60 | 6.82 | 5.55 |
| GPT-oss | 6.36 | 7.35 | 5.27 | 2.77 | 6.04 | 4.55 | 5.02 | 6.68 | 7.83 | 6.54 | 6.74 | 6.11 | 7.43 | 5.34 | 6.94 | 8.04 | 2.65 | 4.83 | 5.79 | 1.03 | 4.46 | 7.56 | 6.41 | 5.28 |
| LLaMA-4 | 3.84 | 6.05 | 4.99 | 2.85 | 5.70 | 4.53 | 4.84 | 4.69 | 7.37 | 6.03 | 5.33 | 5.39 | 5.26 | 4.71 | 6.49 | 6.84 | 1.58 | 4.05 | 5.45 | 1.48 | 3.69 | 4.10 | 4.43 | 4.54 |
| Sarvam-2B | 3.92 | 9.42 | 5.70 | 1.95 | 6.16 | 3.65 | 4.55 | 7.62 | 8.92 | 5.29 | 9.07 | 5.83 | 8.63 | 2.81 | 9.46 | 7.20 | 7.17 | 7.95 | 6.03 | 1.02 | 1.02 | 9.74 | 8.46 | 1.00 |
| Sutra | 8.04 | 8.50 | 5.44 | 1.97 | 6.39 | 5.15 | 4.98 | 7.36 | 8.33 | 7.00 | 8.38 | 5.88 | 8.75 | 5.36 | 8.35 | 8.45 | 7.73 | 8.76 | 6.04 | 6.59 | 3.49 | 9.38 | 8.04 | 5.15 |
| IST | **9.12** | 10.15 | **8.18** | **2.84** | **7.86** | **5.44** | **6.29** | **8.95** | **11.01** | **8.59** | **10.30** | **7.80** | **11.33** | **5.67** | **11.11** | **10.39** | **10.07** | **9.40** | **8.70** | **3.60** | **5.40** | **11.32** | **10.70** | 5.55 |

## 4.2 Extrinsic Evaluation on Downstream Tasks

We also evaluated the downstream model performance (see Table 8) by pretraining LLaMA-3.2 1B models using two tokenizers: i) IndicSuperTokenizer, our proposed tokenizer optimized for morphologically meaningful segmentation in Indic and multilingual settings, and (ii) LLaMA-4 tokenizer, chosen for comparable vocabulary size and widespread use. Both models were trained on the same dataset in iso-compute setting to ensure a fair comparison. More details in the Appendix B. We find that our tokenizer shows competitive performance across the English and Indic benchmarks. We additionally trained a model using the Stage-1 tokenizer, and it attains strong downstream performance. As shown in Table 25 in Appendix, the Stage-1 tokenizer itself constitutes a strong and competitive baseline.

The pretraining corpus (Table 20 in Appendix) balances coverage and domain diversity. It combines web-scale sources (Nemotron CC) for general context with structured data including MegaMath, StackV2, synthetic generations, and books. Indic-language content constitutes roughly 20% of the corpus, drawn from Indic CC, Wikipedia, and Sangraha Verified, providing sufficient signal to evaluate cross-lingual and morphologically rich representation quality.

## 4.3 How does tokenizer design impact model latency and throughput?

Next, we evaluate how tokenization impacts end-to-end model efficiency. We trained two 1B-parameter models under identical conditions: one with our tokenizer and one with the LLaMA tokenizer of similar vocabulary size. We then evaluated inference efficiency over 200 samples spanning Indic languages and English, with varying input lengths. Latency[6] was measured using standard metrics, including Time-To-First-Token (TTFT), Output Throughput (OTPT), and Input Sequence Length (ISL), across 200 instances ( See Appendix C.4 for details) with 5 warm-up requests and results averaged over 10 runs. Experiments were served on 8 H100 GPUs using Triton Inference Server as backend, with a maximum generation limit of 256 new tokens. Our tokenizer yields clear efficiency gains (Table 5). These gains stem from improved compression: shorter token sequences encode more information per token, thereby lowering per-request computation without compromising expressivity. Overall, this demonstrates that tokenizer design directly shapes not only pretraining efficiency but also real-world deployment latency, making it a critical factor for practical model performance.

---

[6] https://tinyurl.com/4e7nh7c8

Table 8: Performance comparison of *English* (left) and *Indic benchmarks* (right).

| English Benchmarks | | | Indic Benchmarks | | |
|---|---|---|---|---|---|
| Dataset | LLaMA-4 | IST | Dataset | LLaMA-4 | IST |
| HellaSwag | 0.353 | 0.357 | Indic COPA | 0.544 | 0.556 |
| CommonsenseQA | 0.206 | 0.204 | Indic Sentiment | 0.524 | 0.551 |
| OpenBookQA | 0.216 | 0.218 | Indic XNLI | 0.347 | 0.346 |
| Winogrande | 0.504 | 0.510 | Indic Paraphrase | 0.534 | 0.539 |
| GSM8K | 0.016 | 0.018 | MILU (Indic Multi-turn LU) | 0.261 | 0.258 |
| ARC Easy | 0.623 | 0.630 | ARC Challenge (Indic) | 0.236 | 0.244 |
| ARC Challenge | 0.291 | 0.292 | TriviaQA (Indic) | 0.268 | 0.262 |
| MMLU | 0.252 | 0.249 | | | |
| DROP | 0.048 | 0.036 | | | |
| Average | 0.279 | 0.279 | Average | 0.388 | **0.394** |

## 4.4 VOCABULARY ALLOCATION: EXPLICIT VS. CORPUS-DRIVEN

We allocate vocabulary budgets proportionally across scripts to preserve subword/multi-word granularity. Budgets were derived from corpus sizes, ensuring that both high- and low-resource scripts retained sufficient capacity. We compared two strategies for realizing this allocation. The first, *explicit merging*, trains script-specific tokenizers and concatenates their vocabularies via a rule-stacking procedure. While conceptually modular, this approach introduces distributional interference across scripts, yielding higher token-to-word ratios and fragmented segmentation (Table 9). The second, *corpus-driven alignment*, trains a single tokenizer on the concatenated multilingual corpus, allowing the vocabulary to adapt naturally to language frequencies. This unified training not only mirrored corpus composition (Table 10) but also achieved the lowest fertility scores across scripts (Table 3), outperforming explicit merging and public baselines. While script-aware budget allocation is necessary, explicit merging is inefficient; corpus-driven alignment provides a more scalable and faithful multilingual tokenization strategy.

Table 9: Fertility comparison between individual script tokenizers and the merged tokenizer across selected Indic languages. Lower values are better.

| Tokenizer | as | bn | hi | mai | mr | san | te |
|---|---|---|---|---|---|---|---|
| Individual | 2.05 | 2.13 | 1.21 | 1.35 | 1.75 | 2.49 | 1.40 |
| Merged | 2.32 | 2.14 | 1.55 | 1.57 | 1.73 | 2.79 | 1.95 |

Table 10: Script-specific training data size (Total corpus size 9.4 GB) and resulting vocabulary percentage distribution. Refer to Table 17 in Appendix for script mapping.

| Metric | ar | bn | deva | en | gu | ka | ml | pa | ta | te |
|---|---|---|---|---|---|---|---|---|---|---|
| Data size (MB) | 106 | 396 | 2200 | 3590 | 124 | 644 | 580 | 307 | 616 | 617 |
| Percentage | 1.12 | 4.18 | 23.25 | 37.94 | 1.31 | 6.81 | 6.13 | 3.24 | 6.51 | 6.52 |
| Vocab perc dist | 2.69 | 6.32 | 20.89 | 32.92 | 2.38 | 7.82 | 6.76 | 4.68 | 7.04 | 8.50 |

## 4.5 QUALITY ANALYSIS: UNDERTRAINED "GLITCH" TOKEN

We analyze under-trained tokens in our tied-embedding LLaMA-3.2-1B models trained with both the IST tokenizer and a comparable BPE tokenizer of similar vocabulary size trained on the same corpus. Both tokenizers share the first 90% of the vocabulary. The IST tokenizer switches to super-word training for the last 10% whereas the base BPE tokeniser continues standard subword training. Following Land & Bartolo (2024) to construct a reference for unused embeddings, we introduced

a small set of dummy tokens into the vocabulary that have zero occurrences in the training data. Their embeddings were averaged to obtain a mean reference vector. We then retrieved the top-$K$ nearest neighbors (cosine distance), which represent potential "glitch" tokens (Geiping et al., 2024). As shown in Figure 5 (in the Appendix) the IST tokenizer produces far fewer such glitch tokens than the base BPE tokenizer. These results suggest that incorporating multi-words promotes more efficient utilization of the vocabulary, while purely subword-based tokenizers overfit in the long tail, yielding a higher proportion of under-trained tokens. More discussion in Appendix Section C.3.

### 4.6 Can we replace Opensource model tokenizer with IST?

Following ReTok (Gu et al., 2024), we replace the tokenizer of a pre-trained LLaMA-3.2-1B model (denoted LLaMA-3.2-ORIG) (Grattafiori et al., 2024) with IndicSuperTokenizer (referred as LLaMA-3.2-IST). Let $V_{\text{orig}}$ and $V_{\text{IST}}$ denote their corresponding vocabularies. For a token $t \in V_{\text{IST}}$, we initialize its embedding $E_{\text{init}}(t)$ as: if $t \in V_{\text{orig}} \cap V_{\text{IST}}$, then $E_{\text{init}}(t) = E_{\text{orig}}(t)$, its embedding from the pretrained model, otherwise, if $t \in V_{\text{IST}} \setminus V_{\text{orig}}$ and decomposes under the original tokenizer into $(t_1, \ldots, t_k)$, then $E_{\text{init}}(t) = \frac{1}{k} \sum_{i=1}^{k} E_{\text{orig}}(t_i)$.

We then continually pretrained the LLaMA-3.2-IST model, keeping just the embedding and LM head layers trainable, on a 40B-token corpus comprising English, Indic, code, and mathematics (see Appendix for details). As seen in Table 11, the LLaMA-3.2-IST model performs competitively with the original LLaMA-3.2-ORIG. This suggests that, in addition to pretraining-from-scratch settings, an optimized multilingual tokenizer, such as IndicSuperTokenizer, could also be leveraged in opensource models through CPT (Continual Pretraining (Chen et al., 2024)) leading to significant throughput gains (as seen in Table 5) while maintaining the original model quality.

Table 11: Performance comparison English (left) and Indic benchmarks (right).

| English Benchmarks | | | Indic Benchmarks | | |
|---|---|---|---|---|---|
| Dataset | LLaMA-3.2-ORIG | LLaMA-3.2-IST | Dataset | LLaMA-3.2-ORIG | LLaMA-3.2-IST |
| Winogrande | 0.60 | 0.61 | Indic COPA | 0.58 | 0.56 |
| GSM8K | 0.05 | 0.05 | Indic Sentiment | 0.82 | 0.85 |
| ARC Challenge | 0.40 | 0.39 | Indic XNLI | 0.35 | 0.34 |
| MMLU | 0.32 | 0.29 | Indic Paraphrase | 0.57 | 0.53 |
| Average | 0.34 | 0.34 | Average | 0.58 | 0.57 |

## 5 Ablation studies

**Two-Stage vs. One-Stage: Controlling Vocabulary** Recently, BoundlessBPE (Schmidt et al., 2024) also explored a one-stage training paradigm in which pre-tokenization is governed by a fixed regular expression, enabling the direct learning of multiword units in a single pass. While effective in capturing frequent expressions, this strategy can also overfit to arbitrary character sequences lacking semantic value, ultimately reducing vocabulary efficiency. Our approach instead introduces a two-stage procedure. We replicate the one-stage setup of BoundlessBPE using its released regex (referred as IST-BR) and compare against our two-stage tokenizer. As shown in Table 12, our method consistently achieves lower fertility across the top 10 Indic languages and English, indicating more compact and semantically grounded vocabularies. Overall, the comparison highlights a clear trade-off: while one-stage methods capture surface-level patterns indiscriminately, our two-stage design balances efficiency and linguistic integrity by decoupling subword and multiword learning.

Table 12: Fertility score ($\downarrow$) comparison between one-stage and two-stage IST tokenizers.

| Tokenizer | as | bn | brx | code | doi | eng | gom | gu | hi | kas | kn | mai | ml | mni | mr | nep | or | pa | san | sat | snd | ta | te | urd |
|---|---|---|---|---|---|---|---|---|---|---|---|---|---|---|---|---|---|---|---|---|---|---|---|---|
| IST-BR (200K) | 1.86 | 1.76 | 2.05 | 1.75 | 1.62 | 1.37 | 2.20 | 1.86 | 1.39 | 1.39 | 2.19 | 1.61 | 2.29 | 2.30 | 1.66 | 1.67 | 1.69 | 1.49 | 2.68 | 3.61 | 1.56 | 2.12 | 1.88 | 1.54 |
| IST (180K/200K) | **1.85** | **1.74** | **2.04** | **1.47** | **1.45** | **1.12** | **2.17** | **1.77** | **1.23** | **1.21** | 2.19 | **1.58** | 2.30 | **2.28** | **1.63** | **1.62** | **1.65** | **1.39** | **2.59** | 3.72 | **1.45** | 2.12 | 1.88 | **1.44** |

**Dataset Size** Similar to (Reddy et al., 2025), we study the effects of scaling training data, however only in Stage 1 of our training. Figure 13 shows that our performance plateaus after 10G of data.

Table 13: Ablation of tokenizer training data size and its impact on fertility score (↓).

| Size | as | bn | brx | code | doi | eng | gom | gu | hi | kas | kn | mai | ml | mni | mr | nep | or | pa | san | sat | snd | ta | te | urd | Average |
|---|---|---|---|---|---|---|---|---|---|---|---|---|---|---|---|---|---|---|---|---|---|---|---|---|---|
| 1G | 3.02 | 2.32 | 2.71 | 1.62 | 1.64 | 1.33 | 1.97 | 1.62 | 1.50 | 1.43 | 2.16 | 1.85 | 2.83 | 2.62 | 1.72 | 2.13 | 1.68 | 1.50 | 2.46 | 13.02 | 1.43 | 1.92 | 1.82 | 1.91 | 2.42 |
| 5G | 1.71 | 1.93 | 2.58 | 1.63 | 1.58 | 1.33 | 2.18 | 1.72 | 1.40 | 1.36 | 2.04 | 1.57 | 2.43 | 2.28 | 1.68 | 1.48 | 1.61 | 1.57 | 2.48 | 4.74 | 1.30 | 2.02 | 1.87 | 1.43 | 1.91 |
| 10G | 1.83 | 1.74 | 1.99 | 1.54 | 1.56 | 1.33 | 2.17 | 1.83 | 1.36 | 1.36 | 2.15 | 1.56 | 2.24 | 2.27 | 1.61 | 1.59 | 1.65 | 1.47 | 2.51 | 3.60 | 1.45 | 2.08 | 1.83 | 1.47 | **1.80** |
| 25G | 1.75 | 1.84 | 2.56 | 1.62 | 1.57 | 1.33 | 2.15 | 1.78 | 1.39 | 1.36 | 2.04 | 1.56 | 2.32 | 2.23 | 1.67 | 1.47 | 1.63 | 1.55 | 2.45 | 3.92 | 1.31 | 2.01 | 1.86 | 1.34 | 1.86 |
| 30G | 1.76 | 1.84 | 2.32 | 1.62 | 1.57 | 1.33 | 2.13 | 1.78 | 1.39 | 1.36 | 2.03 | 1.57 | 2.31 | 2.24 | 1.67 | 1.47 | 1.63 | 1.54 | 2.45 | 4.02 | 1.31 | 2.00 | 1.87 | 1.35 | 1.86 |
| 50G | 1.72 | 1.82 | 2.25 | 1.60 | 1.57 | 1.34 | 2.14 | 1.82 | 1.39 | 1.36 | 2.03 | 1.58 | 2.28 | 2.22 | 1.69 | 1.49 | 1.64 | 1.52 | 2.44 | 4.54 | 1.31 | 2.01 | 1.87 | 1.34 | 1.87 |

**Transition Point**   We ablate the transition point $t$ (Section 3.1) at which training shifts from sub-word to cross-word merges. Varying $t$ reveals a clear trade-off: early transitions favor frequent multiword expressions but weaken morphological coverage, while late transitions preserve subwords at the cost of longer sequences. Across Indic and non-Indic languages, intermediate values of 90% $t$ yield the best balance, improving token efficiency and cross-lingual consistency (Table 14).

Table 14: Impact of varying transition point (as a % of vocab size 200K) on fertility (↓).

| Transition (%) | as | bn | brx | code | doi | eng | gom | gu | hi | kas | kn | mai | ml | mni | mr | nep | or | pa | san | sat | snd | ta | te | urd |
|---|---|---|---|---|---|---|---|---|---|---|---|---|---|---|---|---|---|---|---|---|---|---|---|---|
| 60 | 1.91 | 1.80 | 2.05 | 1.39 | 1.38 | 1.04 | 2.16 | 1.77 | 1.16 | 1.15 | 2.17 | 1.53 | 2.30 | 2.29 | 1.56 | 1.58 | 1.68 | 1.39 | 2.48 | 3.89 | 1.43 | 2.11 | 1.86 | 1.45 |
| 75 | 1.91 | 1.79 | 2.05 | 1.41 | 1.38 | 1.04 | 2.16 | 1.77 | 1.16 | 1.15 | 2.16 | 1.53 | 2.30 | 2.28 | 1.56 | 1.58 | 1.68 | 1.39 | 2.47 | 3.91 | 1.43 | 2.10 | 1.86 | 1.45 |
| 80 | 1.89 | 1.78 | 2.03 | 1.41 | 1.38 | 1.05 | 2.15 | 1.77 | 1.16 | 1.16 | 2.14 | 1.53 | 2.28 | 2.26 | 1.56 | 1.57 | 1.67 | 1.39 | 2.46 | 3.83 | 1.42 | 2.08 | 1.83 | 1.44 |
| 85 | 1.87 | 1.77 | 2.01 | 1.43 | 1.39 | 1.06 | 2.13 | 1.76 | 1.17 | 1.16 | 2.13 | 1.53 | 2.26 | 2.25 | 1.56 | 1.56 | 1.66 | 1.39 | 2.46 | 3.78 | 1.42 | 2.07 | 1.82 | 1.44 |
| 90 | 1.85 | 1.74 | 2.04 | 1.47 | 1.45 | 1.12 | 2.17 | 1.77 | 1.23 | 1.21 | 2.19 | 1.58 | 2.30 | 2.28 | 1.63 | 1.62 | 1.65 | 1.39 | 2.59 | 3.72 | 1.45 | 2.12 | 1.88 | 1.44 |
| 95 | 1.85 | 1.75 | 1.98 | 1.47 | 1.42 | 1.10 | 2.13 | 1.74 | 1.21 | 1.20 | 2.12 | 1.53 | 2.23 | 2.24 | 1.56 | 1.56 | 1.66 | 1.41 | 2.46 | 3.68 | 1.43 | 2.06 | 1.81 | 1.44 |

**Vocabulary Size**   Vocabulary size strongly influences tokenization-model efficiency with trade-offs. Smaller vocabularies yield finer subword units that generalize well to unseen words but lengthen sequences, raising compute costs. Larger vocabularies shorten sequences by encoding frequent forms as single tokens, but waste capacity on rare items, inflate embeddings and softmax layers (Shazeer et al., 2017), and bias toward high-resource languages, hurting multilingual balance. With the same transition point at 90%, we found no significant impact on fertility scores beyond 200K (Table 15).

Table 15: Ablation on vocab size ($t = 90\%$) and its impact on fertility (↓) scores.

| Vocab Size | as | bn | brx | code | doi | eng | gom | gu | hi | kas | kn | mai | ml | mni | mr | nep | or | pa | san | sat | snd | ta | te | urd |
|---|---|---|---|---|---|---|---|---|---|---|---|---|---|---|---|---|---|---|---|---|---|---|---|---|
| 162K/180K | 1.89 | 1.78 | 2.08 | 1.48 | 1.47 | 1.13 | 2.21 | 1.80 | 1.24 | 1.22 | 2.22 | 1.60 | 2.35 | 2.27 | 1.65 | 1.65 | 1.68 | 1.42 | 2.62 | 3.84 | 1.48 | 2.16 | 1.91 | 1.47 |
| 180K/200K | 1.85 | 1.74 | 2.04 | 1.47 | 1.45 | 1.12 | 2.17 | 1.77 | 1.23 | 1.21 | 2.19 | 1.58 | 2.30 | 2.27 | 1.63 | 1.62 | 1.65 | 1.39 | 2.59 | 3.72 | 1.45 | 2.12 | 1.88 | 1.44 |
| 202K/225K | 1.81 | 1.70 | 1.99 | 1.44 | 1.43 | 1.10 | 2.14 | 1.72 | 1.20 | 1.19 | 2.14 | 1.55 | 2.24 | 2.21 | 1.59 | 1.59 | 1.60 | 1.36 | 2.55 | 3.59 | 1.41 | 2.08 | 1.82 | 1.41 |
| 225K/250K | 1.78 | 1.67 | 1.95 | 1.42 | 1.42 | 1.09 | 2.11 | 1.69 | 1.19 | 1.17 | 2.10 | 1.53 | 2.20 | 2.17 | 1.57 | 1.57 | 1.57 | 1.34 | 2.52 | 3.45 | 1.38 | 2.04 | 1.77 | 1.38 |

**Effect of Normalization in Multilingual Tokenization**   Unicode normalization is crucial for multilingual settings (Karthika et al., 2025b), particularly for Indic languages, where a single grapheme can be represented by multiple Unicode sequences (e.g., pre-composed characters vs. base-plus-diacritic sequences), causing token fragmentation and inflated vocabulary size. Table 16 shows that NFKC yielded marginal but consistent gains by unifying character forms. Accordingly, we adopt NFKC to reduce variability and improve tokenizer robustness.

# 6   CONCLUSION

In this work, we revisit tokenization as a central design choice for multilingual LLMs, focusing on Indic languages that expose the limitations of existing subword methods. Our proposed IndicSuper-Tokenizer combines linguistically grounded pre-tokenization with a two-stage subword–superword learning process, yielding more compact and semantically faithful vocabularies. Experiments across intrinsic metrics, downstream task performance, ablations, and inference latency demonstrate consistent gains in efficiency, morphological alignment, and deployment cost, establishing tokenization as a key lever for building equitable and scalable multilingual models.

Table 16: Fertility scores with NFC, NFD, NFKC normalization for all languages.

| Tokenizer | as | bn | brx | code | doi | eng | gom | gu | hi | kas | kn | mai | ml | mni | mr | nep | or | pa | san | sat | snd | ta | te | urd |
|---|---|---|---|---|---|---|---|---|---|---|---|---|---|---|---|---|---|---|---|---|---|---|---|---|
| NFC | 1.8520 | 1.7449 | 2.0412 | 1.4658 | 1.4520 | 1.1167 | 2.1741 | 1.7664 | 1.2250 | 1.2042 | 2.1845 | 1.5761 | 2.3025 | 2.2421 | 1.6273 | 1.6241 | 1.6464 | 1.3915 | 2.5859 | 3.7170 | 1.4515 | 2.1226 | 1.8754 | 1.4371 |
| NFD | 1.8518 | 1.7454 | 2.0413 | 1.4665 | 1.4521 | 1.1168 | 2.1661 | 1.7667 | 1.2252 | 1.2044 | 2.1905 | 1.5765 | 2.3019 | 2.2487 | 1.6274 | 1.6246 | 1.6465 | 1.3917 | 2.5864 | 3.7170 | 1.4523 | 2.1227 | 1.8757 | 1.4377 |
| NFKC | 1.8512 | 1.7430 | 2.0409 | 1.4647 | 1.4520 | 1.1155 | 2.1738 | 1.7644 | 1.2239 | 1.2041 | 2.1812 | 1.5762 | 2.2991 | 2.2327 | 1.6258 | 1.6234 | 1.6420 | 1.3884 | 2.5855 | 3.7172 | 1.4505 | 2.1200 | 1.8724 | 1.4369 |

## ETHICS AND REPRODUCIBILITY STATEMENT

**Ethics Statement**    This work focuses on the responsible development of multilingual tokenization methods for Indian languages. We did not collect or utilize any sensitive or Personally Identifiable Information (PII). All external datasets, libraries, and tools employed in this work are appropriately acknowledged through citations. Since the study did not involve personal, medical, or otherwise sensitive information, formal IRB approval was not required. Throughout the process, we aimed to minimize biases that could disadvantage low-resource languages. We provide our exhaustive study to advance the development of inclusive and efficient multilingual language models.

**Reproducibility Statement**    To promote transparency and reproducibility, we will release the artifacts publicly to benchmark performance of Indian tokenizers, along with detailed documentation. Detailed records of experimental setups, hyperparameters, and evaluation protocols are maintained to allow replication of our results with the implementation details in the Appendix. In addition, we provide ablation studies to facilitate fair benchmarking and enable future research on Indian and multilingual tokenization.

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

# A    APPENDIX

## A.1    LANGUAGE DETAILS

Table 17: Linguistic composition of the 22 scheduled Indian languages analyzed in this work, with their corresponding scripts.

| Family | Script | Languages |
|---|---|---|
| Indo-Aryan | Devanagari | Hindi, Marathi, Maithili, Dogri, Konkani, Sanskrit, Nepali, Kashmiri |
| | Bengali (bn) | Assamese, Bengali |
| | Gurmukhi (pa) | Punjabi |
| | Arabic (ar) | Urdu, Sindhi |
| Dravidian | Kannada (kn) | Kannada |
| | Malayalam (ml) | Malayalam |
| | Tamil (ta) | Tamil |
| | Telugu (te) | Telugu |
| Tibeto-Burman | Devanagari | Bodo |
| | Meitei Mayek | Manipuri (Meitei Mayek script) |
| Austroasiatic | Ol Chiki (sat) | Santali |

Table 18: Mapping of ISO codes to corresponding 22 Indic languages.

| Code | Language | Code | Language | Code | Language |
|---|---|---|---|---|---|
| as | Assamese | bn | Bengali | brx | Bodo |
| doi | Dogri | gu | Gujarati | hi | Hindi |
| kn | Kannada | ks | Kashmiri | gom | Konkani |
| mai | Maithili | ml | Malayalam | mni | Manipuri |
| mr | Marathi | ne | Nepali | or | Odia |
| pa | Punjabi | san | Sanskrit | sat | Santali |
| snd | Sindhi | ta | Tamil | te | Telugu |
| ur | Urdu | | | | |

# B    IMPLEMENTATION

## B.1    TOKENIZER IMPLEMENTATION

We based our training code for the tokenizer on the open implementation of SuperBPE[7] using HuggingFace library (Jain, 2022). We also explored merging tokenizers based on the default priority based BPE in SentencePiece[8]. While we explored implementing the multi-word two stage curriculum in the SentencePiece, we found that it was not trivial. On the other hand, HuggingFace showed issues with the merging strategy. We thus relied on different implementations for different approaches.

## B.2    TRAINING DETAILS

We provide more details about our training setup as discussed in Section 4.2. Each model was trained for 50B tokens under matched hyperparameters (learning rate, batch size, training steps), aligning FLOPs to isolate tokenizer effects. The evaluation was performed using `lm-eval-harness`

---

[7]https://github.com/PythonNut/superbpe/tree/main
[8]https://github.com/google/sentencepiece

(Gao et al., 2024) across standard English benchmarks (MMLU, GSM8K, Winogrande, TriviaQA, HellaSwag, ARC, OpenBookQA, CommonsenseQA, DROP) and Indic benchmarks (IndicCOPA, IndicSentiment, IndicXParaphrase, IndicXNLI (Doddapaneni et al., 2023), ARC Challenge Indic (Sarvam AI, 2025), and MILU Verma et al. (2024)). We report EM for GSM8K and TriviaQA, F1 for DROP, and Accuracy for other benchmarks. Shot settings were fixed per task: 25-shot for ARC/ARC Challenge Indic, 10-shot for HellaSwag, 5-shot for MMLU, GSM8K, and TriviaQA, and zero-shot for the remainder. This setup allows a direct assessment of how tokenizer design influences pretraining efficiency, semantic representation, and generalization across English and Indic tasks.

Table 19: Pretraining configuration for different tokenizers.

| Tokenizer | Architecture | Parameters | Data Size (B) | Learning Rate | Train Steps | Context Length | Batch Size | Vocab Size |
|---|---|---|---|---|---|---|---|---|
| LLaMA-4 | LLaMA-3.2 | 1B | 53.24 | $5 \times 10^{-5}$ | 68000 | 4096 | 192 | 201134 |
| IST | LLaMA-3.2 | 1B | 53.18 | $5 \times 10^{-5}$ | 68000 | 4096 | 192 | 200008 |

Table 20: Pretraining corpus distribution across domains and token count. Indic content is emphasized to reflect multilingual objectives.

| Category | Sources | Percentage (%) | Token Count (B) |
|---|---|---|---|
| Web | Nemotron CC | 30 | 15 |
| Math | MegaMath | 15 | 7.5 |
| Code | StackV2 | 15 | 7.5 |
| Synthetic | New Generations | 10 | 5 |
| Books | Archive | 10 | 5 |
| Indic | Indic CC | 8 | 4 |
| Indic | Indic Wiki | 4 | 2 |
| Indic | Sangraha Verified | 8 | 4 |
| **Total** | | 100 | 50 |

# C  ADDITIONAL DISCUSSION

## C.1  MISMATCH BETWEEN LOSS AND TASK PERFORMANCE

Although toknizers, incorporating multi-word often show slightly higher loss (Liu et al., 2025a) during training compared to models using traditional atomic tokenizers like SentencePiece/BPE, this does not necessarily translate to worse downstream performance. We hypothesize that this is due to two complementary factors. First, the introduction of longer or multi-word tokens such as "to the" or "as well as" increases the number of semantically overlapping candidates, making the model's prediction space less sharply peaked. This means the model may distribute probability across several plausible completions (e.g., "to", "to the", "to be"), thereby lowering the maximum assigned probability to the correct token and inflating the cross-entropy loss. In contrast, other BPE tokenizers often yield only one atomic candidate for such function words, allowing sharper predictions with lower loss. Second, IST tokenizes text into fewer, more meaningful units, so when computing the average loss per token, each mistake contributes more heavily to the total. As a result, although the model learns more compact and generalizable representations, its token-level loss appears higher. This creates a divergence between model loss and real-world task accuracy, indicating that traditional loss curves may underrepresent the representational efficiency and practical utility of compositional tokenizers like IST.

## C.2  MORPHOLOGICALLY GROUNDED TOKEN SPLITTING

We investigate the impact of incorporating morphological information into tokenization for Indic languages (Brahma et al., 2025). The approach involves pre-processing text with a morphology analyzer to segment words into morphemes prior to training. This experiment focuses on languages in the Devanagari script.

We compare two variants: Tokenizer A, trained on raw text, and Tokenizer B, trained on morphologically segmented text using morphology analyzer (Kunchukuttan, 2020). At inference time, Tokenizer B requires the same pre-processing for consistency. Tokenizer B exhibits more semantically

coherent superwords, reflecting meaningful morpheme combinations (Figure 3, 4). This promotes better generalization across related forms and reduces the raw token-to-word ratio, as morpheme-based units are more compressible. Sample outputs (Figures 3, 4) illustrate the contrast between surface-level splits and linguistically aligned segmentations.

Despite these gains, we do not adopt this approach in our final tokenizer. The primary limitation is latency, as the pipeline requires both language identification and morphological analysis. For completeness, we evaluated a Hindi morphology-aware tokenizer augmented with a morphological analyzer (Kunchukuttan, 2020) combined with language identification (LID)[9]. We performed inference on approximately 4-5 MB of Hindi text and measured throughput over 10 runs (with 5 warm-up runs), comparing against our IST tokenizer. Our tokenizer achieved 194K tokens/sec, whereas the morphology-aware tokenizer achieved 90K tokens/sec, representing a 53.28% reduction in throughput. This slowdown arises entirely from the additional LID and morphology-analysis stages, underscoring the efficiency advantages of our approach even when compared to linguistically informed baselines. Extending robust analyzers across all Indic languages also introduces engineering overhead and brittle dependencies. Nevertheless, morphology-aware tokenization remains a promising direction if fast, reliable analyzers become widely available.

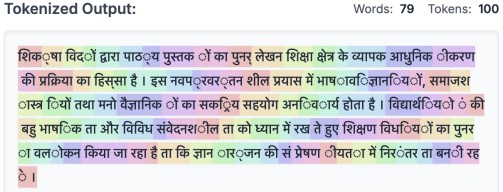

Figure 3: Tokenized output of morph-aware tokenizer

Figure 4: Tokenized output of non morph-aware tokenizer

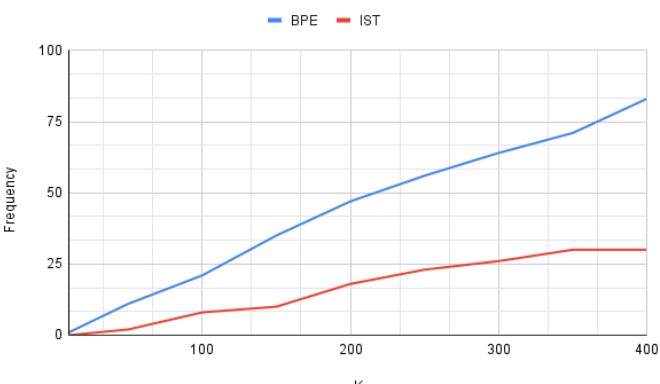

Figure 5: Trend of potential glitch tokens in upper 20K of vocabulary for different K.

## C.3 MORE ON GLITCH TOKENS

For each tokenizer, we vary $K \in \{10, 50, 100, 150, \ldots, 400\}$ to select the top-$K$ embeddings closest to a reference vector derived from artificially unused tokens in the vocabulary (Land & Bartolo, 2024; Geiping et al., 2024). For the IST tokenizer, we count the number of multi-word tokens within the top-$K$. For the BPE variant, we count tokens with IDs $> 180,000$, which corresponds to the upper 20K of the vocabulary. Both tokenizers share the first 180K IDs; the difference lies in how the final 20K IDs are utilized: IST allocates this space for frequent multi-word tokens, while the BPE tokenizer continues learning subwords. This design choice allows IST to more effectively

---

[9]https://pypi.org/project/langdetect/

utilize the tail of the vocabulary for meaningful units, whereas the BPE tokenizer exhibits overfitting in low-frequency subwords. The trend of these counts across different top-$K$ values is visualized in Figure 5. As $K$ increases, the fraction of multi-word tokens in IST remains low but stable, while the BPE variant consistently shows a higher fraction of under-trained subwords, indicating overfitting in the residual vocabulary space.

## C.4 MORE ON LATENCY AND THROUGHPUT EVALUATION

To obtain reliable latency and throughput measurements, we constructed a 200-example multilingual inference set intended to approximate realistic LLM workloads. The set contains diverse sentence-completion style prompts representative of common generation patterns. We include 20 inputs per language across English and nine major Indic languages, ensuring balanced coverage of script diversity, lexical variation, and syntactic complexity. Table 21 presents the token-length distribution of these examples under both the LLaMA-4 tokenizer and our IndicSuperTokenizer, allowing a controlled comparison of inference efficiency across tokenization schemes.

Table 21: Token-length statistics for the 200-example inference set. We report min, p75, p90, p99, maximum, and average token lengths.

| Tokenizer | min | p75 | p90 | p99 | max | avg |
|---|---|---|---|---|---|---|
| Llama-4 | 288 | 805 | 1157 | 2583 | 2869 | 784 |
| IndicSuperTokenizer | 178 | 440 | 541 | 654 | 676 | 379 |

## C.5 DETAILS ABOUT BASELINE TOKENIZERS

Tokenizer fertility is shaped by multiple factors including training data distribution, vocabulary construction, and underlying algorithmic choices, yet publicly available documentation on these aspects is often limited. Table 22 summarizes the vocabulary sizes, training methodologies, and any disclosed data distributions for all baseline tokenizers considered in our study.

Table 22: Summary of baseline tokenizers and publicly available training details.

| Tokenizer | Vocab Size | Training Algorithm / Framework | Data Distribution |
|---|---|---|---|
| DeepSeek-R1 | 128K | BPE (undisclosed variant) | Not publicly disclosed |
| Gemma-3 | 262K | SentencePiece | 140+ languages |
| GPT-OSS | 200K | o200k_harmony (TikToken variant) | Not publicly disclosed |
| LLaMA-3.2-1B | 128K | BPE / SentencePiece-based | Not publicly disclosed |
| LLaMA-4 | 200K | BPE | Not fully disclosed |
| Mistral-Nemo | 131K | Tekken tokenizer (TikToken-based) | 100+ languages; multilingual + code |
| Qwen-3 | 151K | Byte-level BPE | Not publicly disclosed |
| Sarvam | 68K | Not publicly disclosed | Not publicly disclosed |
| Sutra | 256K | SentencePiece (unigram/BPE hybrid) | Balanced multilingual; uniform sampling |

# D METRICS DEFINITIONS

Here, we discuss the different intrinsic metrics used in our evaluation framework.

## D.1 TOKEN-TO-WORD RATIO

The Token-to-word ratio measures the average number of tokens required to represent a single word. It captures the degree of segmentation induced by a tokenizer and is particularly informative for morphologically rich languages where excessive fragmentation increases sequence length. We report this metric to evaluate whether tokenizers balance compact representations with sufficient linguistic coverage.

## D.2 BYTES-PER-TOKEN

Bytes-per-token quantifies the average number of raw text bytes contained in a token. Since scripts differ substantially in character set size and encoding, this metric provides a language-agnostic measure of efficiency. Higher values indicate that tokens encode more information per unit, which reduces sequence length. We include this metric to enable direct comparison of tokenizers across writing systems.

## D.3 NORMALIZED SEQUENCE LENGTH

Normalized sequence length measures the average length of tokenized sequences relative to a chosen base tokenizer. Instead of reporting absolute sequence lengths, this metric highlights how much longer or shorter sequences become when compared to an established reference. It enables fairer cross-tokenizer comparisons since raw lengths can vary significantly across languages and corpora. A normalized value greater than one indicates that the tokenizer produces longer sequences than the baseline, while a value less than one reflects more compact tokenization. We include this metric to directly assess relative efficiency in sequence compression.

## D.4 REYNI'S EFFICIENCY

Rényi's entropy measures the uncertainty of token distributions induced by a tokenizer, extending Shannon entropy by allowing different orders to emphasize frequent or rare tokens. A tokenizer with a very large vocabulary may contain many infrequent tokens that are poorly utilized, while a very small vocabulary forces overuse of common tokens. Entropy therefore reflects how effectively the vocabulary is allocated. To complement this, Rényi's efficiency normalizes entropy with respect to vocabulary size, providing a scale-invariant view of how well the vocabulary capacity is utilized. Together, these metrics characterize both the distributional balance of tokens and the comparative efficiency of different vocabulary scales.

# E EXTENDED RESULTS

In the main paper, due to space constraints, we limited the number of tokenizers presented. Here, we provide an extended list including all of our baseline tokenizers.

Table 23: Comparison of NSL scores (Base LLaMA-4) for different tokenizers across all languages.

| Tokenizer (↓) | as | bn | brx | code | doi | eng | gom | gu | hi | kas | kn | mai | ml | mni | mr | nep | or | pa | san | sat | snd | ta | te | urd |
|---|---|---|---|---|---|---|---|---|---|---|---|---|---|---|---|---|---|---|---|---|---|---|---|---|
| DeepSeek-R1 | 0.83 | 0.97 | 1.25 | 1.03 | 1.29 | 0.98 | 1.28 | 1.48 | 1.59 | 1.29 | 1.41 | 1.34 | 1.52 | 0.99 | 1.49 | 1.61 | 0.67 | 1.41 | 1.19 | 0.69 | 1.34 | 0.82 | 1.34 | 1.21 |
| Gemma-3 | 0.63 | 0.59 | 0.87 | 1.31 | 0.91 | 1.06 | 0.94 | 0.76 | 0.83 | 0.93 | 0.81 | 0.89 | 0.73 | 0.81 | 0.76 | 0.83 | 0.44 | 0.89 | 0.84 | 0.59 | 0.99 | 0.45 | 0.67 | 0.85 |
| GPT-oss | 0.63 | 0.83 | 0.95 | 1.03 | 0.96 | 1.00 | 0.96 | 0.71 | 0.94 | 0.95 | 0.79 | 0.90 | 0.72 | 0.89 | 0.94 | 0.85 | 0.60 | 0.85 | 0.94 | 1.43 | 0.83 | 0.56 | 0.71 | 0.88 |
| LLaMA-3.2-1B | 1.90 | 2.71 | 1.08 | 1.02 | 1.36 | 0.99 | 1.22 | 2.91 | 1.47 | 1.36 | 3.30 | 1.16 | 3.25 | 1.92 | 1.41 | 1.44 | 1.48 | 2.45 | 1.19 | 1.34 | 1.33 | 2.11 | 3.01 | 1.58 |
| LLaMA-4 | 1.00 | 1.00 | 1.00 | 1.00 | 1.00 | 1.00 | 1.00 | 1.00 | 1.00 | 1.00 | 1.00 | 1.00 | 1.00 | 1.00 | 1.00 | 1.00 | 1.00 | 1.00 | 1.00 | 1.00 | 1.00 | 1.00 | 1.00 | 1.00 |
| Mistral-Nemo | 1.00 | 0.95 | 1.06 | 1.15 | 1.07 | 1.06 | 1.09 | 1.09 | 1.12 | 1.08 | 0.91 | 1.08 | 0.95 | 0.95 | 1.13 | 1.21 | 1.57 | 0.98 | 1.04 | 1.34 | 1.20 | 0.63 | 0.64 | 0.95 |
| Qwen-3 | 1.68 | 2.37 | 1.78 | 1.11 | 1.85 | 1.03 | 1.72 | 2.59 | 2.65 | 2.16 | 2.69 | 1.97 | 2.57 | 1.72 | 2.35 | 2.47 | 1.19 | 2.37 | 1.92 | 0.96 | 1.37 | 1.63 | 2.45 | 1.63 |
| Sutra | 0.55 | 0.74 | 0.93 | 2.09 | 0.92 | 0.89 | 0.96 | 0.68 | 0.92 | 0.91 | 0.67 | 0.94 | 0.65 | 0.92 | 0.84 | 0.82 | 0.24 | 0.51 | 0.91 | 0.26 | 1.10 | 0.47 | 0.59 | 0.90 |
| Sarvam | 0.99 | 0.66 | 0.91 | 1.50 | 1.00 | 1.27 | 0.94 | 0.64 | 0.85 | 1.19 | 0.62 | 0.99 | 2.19 | 0.72 | 0.96 | 0.24 | 0.54 | 0.93 | 1.45 | 3.63 | 0.45 | 0.56 | 4.25 |
| IST-BR | 0.45 | 0.61 | 0.66 | 1.28 | 0.89 | 1.04 | 0.84 | 0.57 | 0.77 | 0.91 | 0.54 | 0.80 | 0.50 | 0.94 | 0.63 | 0.69 | 0.18 | 0.48 | 0.70 | 0.45 | 0.78 | 0.38 | 0.44 | 0.92 |
| IST | 0.45 | 0.60 | 0.65 | 0.94 | 0.78 | 0.85 | 0.82 | 0.54 | 0.68 | 0.80 | 0.53 | 0.76 | 0.50 | 0.91 | 0.61 | 0.67 | 0.18 | 0.45 | 0.66 | 0.45 | 0.72 | 0.38 | 0.44 | 0.86 |

Table 24: Fertility scores across tokenizers and languages. Lower is better.

| Tokenizer (↓) | as | bn | brx | code | doi | eng | gom | gu | hi | kas | kn | mai | ml | mni | mr | nep | or | pa | san | sat | snd | ta | te | urd |
|---|---|---|---|---|---|---|---|---|---|---|---|---|---|---|---|---|---|---|---|---|---|---|---|---|
| DeepSeek-R1 | 3.54 | 2.88 | 4.23 | 1.53 | 2.66 | 1.34 | 3.68 | 4.92 | 3.02 | 2.49 | 6.01 | 3.21 | 7.95 | 2.67 | 4.17 | 3.97 | 7.13 | 4.48 | 5.07 | 6.12 | 2.82 | 4.92 | 6.13 | 2.17 |
| Gemma3 | 2.65 | 1.69 | 2.84 | 1.79 | 1.69 | 1.39 | 2.60 | 2.50 | 1.47 | 1.48 | 3.34 | 1.91 | 3.45 | 2.07 | 2.03 | 2.03 | 4.42 | 2.83 | 3.37 | 5.16 | 2.03 | 2.50 | 2.94 | 1.44 |
| GPT-OSS | 2.66 | 2.41 | 3.17 | 1.51 | 1.89 | 1.33 | 2.73 | 2.37 | 1.72 | 1.58 | 3.34 | 2.01 | 3.51 | 2.41 | 2.61 | 2.10 | 6.26 | 2.71 | 3.89 | 13.01 | 1.76 | 3.18 | 3.13 | 1.51 |
| Llama-3.2-1B | 8.44 | 8.08 | 3.64 | 1.51 | 2.92 | 1.35 | 3.46 | 9.95 | 2.74 | 2.70 | 14.44 | 2.79 | 16.26 | 5.31 | 3.90 | 3.52 | 15.68 | 7.88 | 4.86 | 12.15 | 2.85 | 12.25 | 13.68 | 2.73 |
| LLaMA-4 | 4.40 | 2.93 | 3.34 | 1.46 | 2.00 | 1.34 | 2.84 | 3.37 | 1.83 | 1.72 | 4.23 | 2.28 | 4.95 | 2.73 | 2.79 | 2.46 | 10.51 | 3.23 | 4.12 | 9.04 | 2.13 | 5.87 | 4.53 | 1.76 |
| Mistral-Nemo | 4.28 | 2.82 | 3.52 | 1.75 | 2.12 | 1.41 | 3.08 | 3.63 | 2.05 | 1.82 | 3.84 | 2.48 | 4.82 | 2.67 | 3.10 | 2.97 | 16.92 | 3.04 | 4.34 | 12.16 | 2.51 | 3.67 | 3.71 | 1.65 |
| Qwen3-32B | 7.47 | 7.11 | 6.10 | 1.68 | 4.05 | 1.41 | 5.08 | 8.87 | 4.86 | 3.70 | 11.48 | 4.53 | 12.77 | 4.76 | 6.56 | 6.10 | 12.37 | 7.60 | 8.04 | 8.81 | 2.95 | 9.69 | 11.10 | 2.90 |
| Sarvam-2B | 4.24 | 1.91 | 2.92 | 2.14 | 1.85 | 1.66 | 3.01 | 2.11 | 1.53 | 1.91 | 2.53 | 2.11 | 3.19 | 4.60 | 1.94 | 2.35 | 2.43 | 1.67 | 3.78 | 13.07 | 7.62 | 2.49 | 2.63 | 7.93 |
| Sutra | 2.12 | 2.07 | 3.06 | 2.12 | 1.78 | 1.17 | 2.68 | 2.15 | 1.62 | 1.48 | 2.71 | 2.08 | 3.10 | 2.40 | 2.18 | 2.01 | 2.24 | 1.50 | 3.76 | 2.03 | 2.23 | 2.58 | 2.77 | 1.55 |
| IST-BR | 1.86 | 1.76 | 2.05 | 1.75 | 1.62 | 1.37 | 2.20 | 1.86 | 1.39 | 1.39 | 2.19 | 1.61 | 2.29 | 2.30 | 1.66 | 1.67 | 1.69 | 1.49 | 2.68 | 3.61 | 1.56 | 2.12 | 1.88 | 1.54 |
| IST | 1.85 | 1.74 | 2.04 | 1.47 | 1.45 | 1.12 | 2.17 | 1.77 | 1.23 | 1.21 | 2.19 | 1.58 | 2.30 | 2.28 | 1.63 | 1.62 | 1.65 | 1.39 | 2.59 | 3.72 | 1.45 | 2.12 | 1.88 | 1.44 |

Table 25: Comparison of downstream performance between IST (Stage-1) and IST (Stage-2).

| English Benchmarks | | | Indic Benchmarks | | |
|---|---|---|---|---|---|
| Dataset | IST-Stage-1 | IST-Stage-2 | Dataset | IST-Stage-1 | IST-Stage-1 |
| HellaSwag | 0.348 | 0.357 | Indic COPA | 0.556 | 0.556 |
| CommonsenseQA | 0.193 | 0.204 | Indic Sentiment | 0.557 | 0.551 |
| OpenBookQA | 0.214 | 0.218 | Indic XNLI | 0.366 | 0.346 |
| Winogrande | 0.515 | 0.510 | Indic Paraphrase | 0.562 | 0.539 |
| GSM8K | 0.021 | 0.018 | MILU (Indic Multi-turn LU) | 0.265 | 0.258 |
| ARC Easy | 0.625 | 0.630 | ARC Challenge (Indic) | 0.247 | 0.244 |
| ARC Challenge | 0.279 | 0.292 | TriviaQA (Indic) | 0.268 | 0.262 |
| MMLU | 0.255 | 0.249 | | | |
| DROP | 0.042 | 0.036 | | | |
| Average | 0.277 | 0.279 | Average | 0.403 | **0.394** |