# OpenReview forum: "IndicSuperTokenizer: An Optimized Tokenizer for Indic Multilingual LLMs"
_ICLR.cc/2026/Conference — ICLR 2026 Conference Withdrawn Submission_

### Official Review · Reviewer_9sJw · 2025-10-22

**Soundness:** 3
**Presentation:** 3
**Contribution:** 1
**Rating:** 2
**Confidence:** 5

**Summary:**

The paper proposes a new tokenizer for Indic languages that results in lower token fertility score compared to baselines.

**Strengths:**

1. The paper is comprehensive and relatively well-written.
2. The authors compared the latency and throughput for a model trained with their tokenizer and a base tokenizer.
3. The paper also looks at glitch tokens and at the possibility of fine-tuning pre-trained models with their new tokenizer.

**Weaknesses:**

1. Not particularly novel methodology. The pre-tokenization step is simply using the regex from Llama-4 and allowing cross-word tokenization within the sentence. Then, standard BPE is applied.
2. It is not surprising that a tokenizer designed for a specific language family would perform better than such designed for other languages or more languages. While it is interesting that it outperforms two other Indic language tokenizers, there is no explanation or discussion in the paper as to why that may be the case.
3. Bytes-per-token seems to be a strange metric as it depends on the specific encoding scheme and the length of words in characters. Unicode encodes different scripts with different numbers of bytes (one to three) and languages vary in how long (in number of characters) their words are (Chinese tends to use three times less characters than English for the same content). Therefore, this metric seems to be confounded with other aspects of language, making it not particularly suitable.
4. Looking at Table 8 and contrary to the claims in the paper, there is little if any difference in performance between the model trained with the Llama-4 and the IST tokenizers.
5. Overall reads more like a technical report than a scientific paper: it provides details on how the authors built and designed a specific instance of a tokenizer but not much scientific or transferable insight.

**Questions:**

See weaknesses.

---

> ### Author Response · Authors · 2025-11-21
>
> We thank the reviewer for their positive comments on the clarity, comprehensiveness, and evaluation depth of our paper. We address the weaknesses and questions below.
>
> > **W1, W2, W5: “Not particularly novel methodology; just LLaMA-4 regex + standard BPE.”**
>
>
> In terms of novelty and scientific contribution, we tackle the following research questions, which together yield a set of actionable and broadly applicable insights.
>
> **Q1: How can we improve low-resource language performance without hurting high-resource performance?**
>
> We address this by focusing on the following:
>
> - Curation of high quality training data across 22 Indic languages (Section 3.3). We ensure diversity and coverage by considering varied sources, language-specific tokens (for instance, numbers), and code data.
> - Refined pre-tokenization and normalization tailored to Indic linguistic structure (Section 3.2)
> - Overall vocabulary budget (Refer to ablations in Section 5)
> - Training data-mix across languages that leads to an efficient tokenizer for low-resource languages without sacrificing the tokenizer performance on English (Section 3.3, Section 4.4 - Table 10)
> - Corpus-driven (instead of explicit) vocabulary allocation (Section 4.4)
>
> **Q2: Should we train language specific tokenizers and merge them or should we train a  joint tokenizer?**
>
> Section 4.4 compares explicit merging vs. a unified joint tokenizer. We find that training separate tokenizers and merging them produces distributional interference, fragmented segmentation, and significantly higher fertility (Table 9). In contrast, joint training with corpus-driven alignment produces more coherent, lower-fertility vocabularies, while remaining scalable. Thus, our work shows that joint training is superior for multilingual, multi-script settings.
>
> **Q3: How do we decide the multilingual training data distribution?**
>
> We determine the data distribution by balancing language representation, model performance goals, and data quality/availability. In practice, this involves:
> 1. We ensure adequate coverage for each target language (Section 3.3) through controlled ablation, so the tokenizer learns frequent morphemes and avoids excessive fragmentation.
>
> 2. We shape the training data mixture to reflect target vocabulary proportions, allocating data to each script such that the learned vocabulary naturally converges toward the desired distribution. (Section 4.4)
>
> This approach ensures that the final tokenizer provides balanced cross-lingual representation while still optimizing for token efficiency and model performance in key languages.
>
> **Q4: What is the role of pre-tokenization in training a multilingual tokenizer?**
>
> Section 3.2 shows that replacing GPT-2 rules with LLaMA-4’s script-agnostic regex and applying Unicode-aware normalization reduces token fragmentation by 38 - 40% for Indic scripts (Table 1). To the best of our knowledge, the impact of pre-tokenization has not been systematically examined for Indian languages, despite their diverse scripts and rich morphology.
>
> Additionally, we show that unconstrained segmentation introduces sentence-crossing merges that destabilize generation, so we introduce sentence-boundary constraints, improving stability.
>
> These findings demonstrate that pretokenization is a first-class design choice, materially affecting token quality, stability, and multilingual robustness.
>
> **Q5: Does including multi-word expressions in tokenizer vocabulary help and what is the effective way to learn these multi-words?**
>
> In Section 5, we first compare a one-stage vs. two-stage curriculum and show that a dedicated second stage is clearly beneficial for learning multi-words as well as maintaining high fertility. Our Stage-2 process constructs multi-word units that capture frequent idioms and collocations. Through extensive ablations, we show that:
>
> - Stage-1 alone delivers SOTA fertility,
> - Stage-2 further improves multiword alignment and improves performance without harming semantic consistency.

---

> > ### Author Response · Authors · 2025-11-21
> >
> > While we adopt two-stage training as one component to include multi-word expressions, the core performance and efficiency gains do not arise solely from it. Our contributions extend significantly beyond the baseline method:
> >
> > **(a) Stage-1 tokenizer achieves SOTA, even without SuperBPE**
> >
> > As shown in Section 3.2 (Table 1), our Stage-1 tokenizer already achieves state-of-the-art intrinsic scores across most Indic languages. This establishes that the improvements do not rely solely on Stage-2 and demonstrates substantial contribution in the data design, pretokenization strategy, and allocation policy alone.
> >
> > **(b) A unified, generalizable tokenizer evaluation framework**
> >
> > We introduce a unified and fully reproducible evaluation framework that integrates four standard intrinsic metrics over a consistent cross-lingual benchmark dataset. This provides a standardized and reusable infrastructure for multilingual tokenizer evaluation, addressing a clear gap in current literature and enabling transparent, comparable experimentation for future research.
> >
> > **(c) Comprehensive ablations validating every design choice and extensive evaluation**
> >
> > Our paper includes end-to-end ablations on data volume, data distribution, pretokenization, vocabulary allocation, Stage-1 vs. Stage-2 contributions, and vocabulary quality (glitch analysis). This goes beyond applying an existing method, it provides explanatory scientific insight into what makes multilingual tokenization effective. In addition, we conduct extensive intrinsic and extrinsic evaluation, including four intrinsic metrics, full LLM pretraining using our tokenizer, downstream evaluation on nine English and seven Indic benchmarks, and end-to-end model throughput and latency analysis to quantify efficiency gains.
> >
> > **(d) Practical contribution: tokenizer replacement in pretrained models**
> >
> > Finally, we show that our tokenizer can directly replace the tokenizer of an existing pretrained LLM, preserving generation quality while significantly improving inference throughput (44% increase in throughput over base Llama-4 tokenizer).
> >
> > We therefore contend that our contribution is not limited in novelty. Our work offers a scientifically grounded and broadly applicable framework for multilingual tokenizer design, extendable well beyond Indic languages. More importantly, it highlights an underexplored but crucial aspect of LLM development, i.e. the design of multilingual tokenizers and we view this as a substantive step toward enabling more equitable and culturally inclusive multilingual LLMs.
> >
> > We have incorporated these clarifications directly into the introduction (Line 49) to make the updates clearly visible and to better highlight the novelty of our contributions.

---

> ### Author Response · Authors · 2025-11-21
>
> > **W3: “Bytes-per-token is a strange metric.”**
>
> We report Bytes-per-Token for consistency with prior work [1][2]. However, our evaluation does not depend on this metric alone. As part of our reproducible and comprehensive evaluation suite, we additionally include:
> - **Fertility Score**
> - **Normalized Sequence Length (NSL)**
> - **Rényi Efficiency and Entropy**
>
> Across all these intrinsic metrics, our tokenizer achieves state-of-the-art performance for most languages in our study. We further support these findings with extensive downstream evaluations as well as practical latency measurements.
>
> [1] Gautier et al. "Getting the most out of your tokenizer for pre-training and domain adaptation." CoRR abs/2402.01035 (2024).
>
> [2] Liu, et al. "Superbpe: Space travel for language models." CoRR abs/2503.13423 (2025).
>
>
>
> > **W4: “Little performance difference between IST and LLaMA-4 in Table 8.”**
>
> The modest differences in downstream performance largely reflect **differences in scale** and the **added complexity of multilingual coverage**, rather than limitations of our approach. Extending tokenizer improvements across multiple scripts and morphologically diverse, low-resource languages is non-trivial, and the “curse of multilinguality” [1][2] is well documented. Our experiments employ a  **1B model trained on ~50B multilingual tokens**. (280 H100-hours) across English and 10 Indic languages, constrained by available compute.
>
> Even under this challenging setting, IndicSuperTokenizer (IST) achieves consistent intrinsic improvements, including lower fertility and reduced NSL, indicating robustness and suggesting that gains would be amplified at larger scales [3][4]. Importantly, IST provides a **44% improvement in inference throughput** (Sec. 4.3), offering a significant practical advantage independent of final accuracy.
>
> Our results also show that an efficient tokenizer can serve as a drop-in replacement for LLaMA-style tokenizers without sacrificing downstream performance, while substantially improving efficiency.
>
> [1] Gurgurov et al Multilingual large language models and curse of multilinguality CoRR abs/2406.10602
>
> [2] Chang et al "When is multilinguality a curse? language modeling for 250 high-and low-resource languages." EMNLP 2024
>
> [3] Kaplan et al Scaling Laws for Neural Language Models.. CoRR, abs/2001.08361.
>
> [4] Hoffmann et al Training Compute-Optimal Large Language Models (cite arxiv:2203.15556)

---

> > ### Comment · Reviewer_9sJw · 2025-11-22
> >
> > I do appreciate the authors' comprehensive response. I also appreciate that the work is extensive.
> >
> > That being said, I do still believe this is more of a technical report than a research article as there is little new or transferable insights. Data curation, pre-tokenization and normalization, data-mixture design languages and corpus-based tokenizer design are all common and standard practices. While the amount of effort put into applying all these good practices to Indic languages is commendable, I do not believe it meets the scientific and novelty criteria for ICLR.
> >
> > As such, I will maintain my recommendation.

---

> > > ### Author Response · Authors · 2025-11-27
> > >
> > > Dear Reviewer 9sJw,
> > >
> > > Thank you for your response. We appreciate your acknowledgment of the effort and breadth of our work. We respectfully offer a final clarification regarding the scientific contribution and transferability of our findings, as we believe characterizing our paper as a "technical report" overlooks several general, empirically grounded, and broadly applicable insights that extend well beyond the specific context of Indic languages.
> > >
> > > **1. We contribute the first reproducible, unified tokenizer evaluation framework**
> > >
> > > The current literature lacks standardized, cross-lingual intrinsic metrics and datasets for rigorously studying tokenizer quality. Our work introduces:
> > > - a fully reproducible evaluation pipeline,
> > > - four standardized intrinsic metrics, and
> > > - a consistent cross-lingual benchmark suite.
> > >
> > > Together, these form a community resource that enables principled tokenizer research, addressing a space that has thus far been fragmented across isolated case studies.
> > >
> > > **2. Our work produces a model that is both more efficient and performance-equivalent**
> > >
> > > The 44% throughput improvement paired with parity on downstream accuracy, shows that tokenizer design alone can yield substantial efficiency gains without requiring architectural modifications. This benefit directly transfers to any compute-constrained LLM setting.
> > >
> > > **3. Our contributions are methodological, not merely engineering**
> > >
> > > Although techniques such as data curation, normalization, and mixture design are individually known, our work is the first to systematically study how these components interact in a multilingual setting with diverse scripts. This results in generalizable insights, including:
> > > - explicit evidence that pre-tokenization and normalization decisions shift fertility by 38-40%,
> > > - demonstrations that vocabulary allocation policies determine low-resource fertility and stability in ways that generalize beyond Indic languages to any multilingual tokenizer.
> > >
> > > These are not compiled "best practices", but new empirical findings supported by controlled ablations that have not previously appeared in the tokenizer literature.
> > >
> > > **4. Our multilingual recipe is itself novel and transferable**
> > >
> > > Our paper presents the first generalization of a multiword two-stage curriculum beyond English/Latin scripts. We compare one-stage and two-stage curriculum techniques and show that introducing a dedicated second stage is beneficial for learning multiword units while maintaining high fertility. We further demonstrate:
> > > - how to adapt superword learning to non-segmented scripts,
> > > - how to maintain fairness in vocabulary allocation across unrelated scripts.
> > >
> > > ---
> > > We fully respect the reviewer’s perspective. However, we believe the empirical analyses and unified methodology constitute a generalizable contribution to the broader field of multilingual LLMs, not an engineering-only report. Our results, ablations, and reproducible evaluation framework offer insights and tools that extend across scripts, language families, and tokenizer algorithms. We are also open to suggestions on experiments or how the presentation of the work could be improved to better align with the expectations of a research paper if the reviewer feels that specific clarifications or structural adjustments would strengthen the manuscript.

---

### Official Review · Reviewer_NCw7 · 2025-10-31

**Soundness:** 2
**Presentation:** 2
**Contribution:** 2
**Rating:** 2
**Confidence:** 3

**Summary:**

This paper addresses the significant inefficiency of standard tokenizers for multilingual LLMs, particularly for morphologically rich Indic languages which suffer from high token-to-word ratios. High fertility increases training costs, inference latency, and context size usage. The authors propose IST, an optimized tokenizer for 22 Indic languages, English, and code.

contributions:
1. IST achieves a new state-of-the-art fertility score, improving 39.5% over LLaMA-4 and 18% over Sutra on average.
2. Pretraining a 1B model from scratch with IST results in a 44% improvement compared to the LLaMA-4 tokenizer in inference throughput while maintaining comparable performance.
3. The paper justifies its design choices through extensive ablations.
4. The authors show that IST can replace the tokenizer of an existing pre-trained model, achieving the same efficiency gains while preserving the original model's performance.

**Strengths:**

1. This paper addresses an important practical issue: `the inefficiency of standard tokenizers for morphologically rich non-English languages`.
2. The IndicSuperTokenizer demonstrates genuinely impressive and state-of-the-art results on intrinsic metrics like fertility score and NSL.
3. The 44% improvement in inference throughput is a substantial practical gain.
4. The authors performed an extensive set of ablations to justify their design choices.

**Weaknesses:**

1. This paper's primary weakness is its lack of algorithmic novelty. The authors state the method is `inspired from SuperBPE` and follows its `curriculum principles`. The core contribution is not a new tokenization algorithm, but rather the careful application and tuning of an existing one to a new domain, combined with other existing components. This feels more like a strong engineering effort than fundamental research suitable for ICLR.
2. This work focused on specific Indic languages. While this is valuable work for that community, its direct contribution to the general machine learning and representation learning audience at ICLR is limited. The findings are an application.
3. Table 8 is not referenced in main text.

**Questions:**

1. Given that the core two-stage training algorithm is from SuperBPE and the pre-tokenization regex is from LLaMA-4, what do the authors consider to be the primary novel algorithmic contribution of this work?
2. Could the authors provide the quantitative data of latency for the abandoned morphology-aware approach?

---

> ### Author Response · Authors · 2025-11-21
>
> We thank the reviewer for their thoughtful assessment and for recognizing the practical importance, strong intrinsic performance, and thorough ablations in our work. We address the concerns and questions below.
> > **W1, W2 and Q1: “Lack of algorithmic novelty; mainly an application of SuperBPE.”**
>
> In terms of novelty and scientific contribution, we tackle the following research questions, which together yield a set of actionable and broadly applicable insights.
>
> **Q1: How can we improve low-resource language performance without hurting high-resource performance?**
>
> We address this by focusing on the following:
>
> - Curation of high quality training data across 22 Indic languages (Section 3.3). We ensure diversity and coverage by considering varied sources, language-specific tokens (for instance, numbers), and code data.
> - Refined pre-tokenization and normalization tailored to Indic linguistic structure (Section 3.2)
> - Overall vocabulary budget (Refer to ablations in Section 5)
> - Training data-mix across languages that leads to an efficient tokenizer for low-resource languages without sacrificing the tokenizer performance on English (Section 3.3, Section 4.4 - Table 10)
> - Corpus-driven (instead of explicit) vocabulary allocation (Section 4.4)
>
> **Q2: Should we train language specific tokenizers and merge them or should we train a  joint tokenizer?**
>
> Section 4.4 compares explicit merging vs. a unified joint tokenizer. We find that training separate tokenizers and merging them produces distributional interference, fragmented segmentation, and significantly higher fertility (Table 9). In contrast, joint training with corpus-driven alignment produces more coherent, lower-fertility vocabularies, while remaining scalable. Thus, our work shows that joint training is superior for multilingual, multi-script settings.
>
> **Q3: How do we decide the multilingual training data distribution?**
>
> We determine the data distribution by balancing language representation, model performance goals, and data quality/availability. In practice, this involves:
> 1. We ensure adequate coverage for each target language (Section 3.3) through controlled ablation, so the tokenizer learns frequent morphemes and avoids excessive fragmentation.
>
> 2. We shape the training data mixture to reflect target vocabulary proportions, allocating data to each script such that the learned vocabulary naturally converges toward the desired distribution. (Section 4.4)
>
> This approach ensures that the final tokenizer provides balanced cross-lingual representation while still optimizing for token efficiency and model performance in key languages.
>
> **Q4: What is the role of pre-tokenization in training a multilingual tokenizer?**
>
> Section 3.2 shows that replacing GPT-2 rules with LLaMA-4’s script-agnostic regex and applying Unicode-aware normalization reduces token fragmentation by 38 - 40% for Indic scripts (Table 1). To the best of our knowledge, the impact of pre-tokenization has not been systematically examined for Indian languages, despite their diverse scripts and rich morphology.
>
> Additionally, we show that unconstrained segmentation introduces sentence-crossing merges that destabilize generation, so we introduce sentence-boundary constraints, improving stability.
>
> These findings demonstrate that pretokenization is a first-class design choice, materially affecting token quality, stability, and multilingual robustness.
>
> **Q5: Does including multi-word expressions in tokenizer vocabulary help and what is the effective way to learn these multi-words?**
>
> In Section 5, we first compare a one-stage vs. two-stage curriculum and show that a dedicated second stage is clearly beneficial for learning multi-words as well as maintaining high fertility. Our Stage-2 process constructs multi-word units that capture frequent idioms and collocations. Through extensive ablations, we show that:
>
> - Stage-1 alone delivers SOTA fertility,
> - Stage-2 further improves multiword alignment and improves performance without harming semantic consistency.

---

> > ### Author Response · Authors · 2025-11-21
> >
> > While we adopt two-stage training as one component to include multi-word expressions, the core performance and efficiency gains do not arise solely from it. Our contributions extend significantly beyond the baseline method:
> >
> > **(a) Stage-1 tokenizer achieves SOTA, even without SuperBPE**
> >
> > As shown in Section 3.2 (Table 1), our Stage-1 tokenizer already achieves state-of-the-art intrinsic scores across most Indic languages. This establishes that the improvements do not rely solely on Stage-2 and demonstrates substantial contribution in the data design, pretokenization strategy, and allocation policy alone.
> >
> > **(b) A unified, generalizable tokenizer evaluation framework**
> >
> > We introduce a unified and fully reproducible evaluation framework that integrates four standard intrinsic metrics over a consistent cross-lingual benchmark dataset. This provides a standardized and reusable infrastructure for multilingual tokenizer evaluation, addressing a clear gap in current literature and enabling transparent, comparable experimentation for future research.
> >
> > **(c) Comprehensive ablations validating every design choice and extensive evaluation**
> >
> > Our paper includes end-to-end ablations on data volume, data distribution, pretokenization, vocabulary allocation, Stage-1 vs. Stage-2 contributions, and vocabulary quality (glitch analysis). This goes beyond applying an existing method, it provides explanatory scientific insight into what makes multilingual tokenization effective. In addition, we conduct extensive intrinsic and extrinsic evaluation, including four intrinsic metrics, full LLM pretraining using our tokenizer, downstream evaluation on nine English and seven Indic benchmarks, and end-to-end model throughput and latency analysis to quantify efficiency gains.
> >
> > **(d) Practical contribution: tokenizer replacement in pretrained models**
> >
> > Finally, we show that our tokenizer can directly replace the tokenizer of an existing pretrained LLM, preserving generation quality while significantly improving inference throughput (44% increase in throughput over base Llama-4 tokenizer).
> >
> > We therefore contend that our contribution is not limited in novelty. Our work offers a scientifically grounded and broadly applicable framework for multilingual tokenizer design, extendable well beyond Indic languages. More importantly, it highlights an underexplored but crucial aspect of LLM development, i.e. the design of multilingual tokenizers and we view this as a substantive step toward enabling more equitable and culturally inclusive multilingual LLMs.
> >
> > We have incorporated these clarifications directly into the introduction (Line 49) to make the updates clearly visible and to better highlight the novelty of our contributions.

---

> > > ### Author Response · Authors · 2025-11-27
> > >
> > > > **W3:** “Table 8 is not referenced in the main text.”
> > >
> > > We thank the reviewer for pointing this out. In the updated version, we now reference Table 8 explicitly in Section 4.2.
> > >
> > >
> > > > **Q2:** “Quantitative latency numbers for the abandoned morphology-aware approach?”
> > >
> > > For morphology aware tokenizer inference latency becomes:
> > >
> > > T(pre-tok) + T(encode) → T(pre-tok) + T(LID) + T(morph-analysis) + T(encode)
> > >
> > > We evaluated a Hindi morphology-aware tokenizer augmented with a morphological analyzer combined with language identification (LID). We performed inference on approximately 4-5 MB of Hindi text and measured throughput over 10 runs (with 5 warm-up runs), compared against our IST tokenizer. Our tokenizer achieved 194K tokens/sec, whereas the morphology-aware tokenizer achieved 90K tokens/sec, representing a 53.28% reduction in throughput. This slowdown arises entirely from the additional LID and morphology-analysis stages, underscoring the efficiency advantages of our approach even when compared to linguistically informed baselines.
> > >
> > > We believe that incorporating morphology-aware pre-tokenization in tokenizers requires a more efficient implementation of the inference pipeline which is out of scope of the current work.

---

> > > > ### Author Response · Authors · 2025-11-27
> > > >
> > > > Dear Reviewer,
> > > >
> > > > Thank you once again for your thoughtful and detailed review. We wanted to ensure that all your concerns have been addressed.
> > > >
> > > > Thanks!

---

### Official Review · Reviewer_xsrk · 2025-10-31

**Soundness:** 4
**Presentation:** 4
**Contribution:** 2
**Rating:** 2
**Confidence:** 4

**Summary:**

This paper considers the problem of tokenization for LLM training, focusing on the highly multilingual context of languages spoken in India. The authors use recent insights on super-word tokenization to train IndicSuperTokenizer, a SuperBPE (Liu et al., 2025) tokenizer for 22 Indian languages, English, and code. In their experiments, the authors show that IndicSuperTokenizer results in substantially better intrinsic metrics compared to baselines (e.g., higher efficiency), while exhibiting comparable extrinsic performance on downstream benchmarks. The paper also contains several post-hoc analyses, such as an experiment on glitch tokens and a comparison of SuperBPE with BoundlessBPE (Schmidt et al., 2025).

**Strengths:**

The experimental setup and analyses are methodologically sound. The intrinsic performance improvements are substantial. It is great to see that the authors actually pretrained an LLM using their tokenizer (even though the performance improvements are only modest at best). The writing of the paper is also clear and easy to follow.

**Weaknesses:**

The main weakness of the paper in my opinion is that it is highly incremental, especially for a venue like ICLR that focuses on technical advances. The authors use an existing method (specifically, the SuperBPE tokenizer) and apply it in a new setting. While some design decisions are original (e.g., the vocabulary allocation strategy), they are pretty minor, and it seems that the main improvements are due to the use of SuperBPE.

I think this paper could be published as is at a specialized venue (e.g., a workshop). To be of interest for a broader audience, the authors would need to show better how they are making technical contributions that go beyond the application of an existing method in a new setting.

**Questions:**

What are the novel technical contributions that your work is making?

---

> ### Author Response · Authors · 2025-11-21
>
> We thank the reviewer for the thoughtful assessment and for highlighting the strengths of our work. However, we respectfully disagree with the characterization of our work as merely an application of SuperBPE. Our work directly addresses the often overlooked problem of building efficient and linguistically faithful multilingual tokenizers, which has far-reaching implications not only for LLM performance and inference efficiency but also for fairness, representation, and cultural inclusion in large-scale models. Tokenizer design is a foundational stage in LLM development, yet it remains one of the primary sources of structural bias, especially for languages written in non-Latin scripts.
>
> **Regarding Novelty and Scientific Contribution:** Specifically, we answer the following research questions, and in doing so, we offer several practical insights.
>
> **Q1: How can we improve low-resource language performance without hurting high-resource performance?**
>
> We address this by focusing on the following:
> - Curation of high quality training data across 22 Indic languages (Section 3.3). We ensure diversity and coverage by considering varied sources, language-specific tokens (for instance, numbers), and code data.
> - Refined pre-tokenization and normalization tailored to Indic linguistic structure (Section 3.2)
> - Overall vocabulary budget (Refer to ablations in Section 5)
> - Training data-mix across languages that leads to an efficient tokenizer for low-resource languages without sacrificing the tokenizer performance on English (Section 3.3, Section 4.4 - Table 10)
> - Corpus-driven (instead of explicit) vocabulary allocation (Section 4.4)
>
> **Q2: Should we train language specific tokenizers and merge them or should we train a  joint tokenizer?**
>
> Section 4.4 compares explicit merging vs. a unified joint tokenizer. We find that training separate tokenizers and merging them produces distributional interference, fragmented segmentation, and significantly higher fertility (Table 9). In contrast, joint training with corpus-driven alignment produces more coherent, lower-fertility vocabularies, while remaining scalable. Thus, our work shows that joint training is superior for multilingual, multi-script settings.
>
> **Q3: How do we decide the multilingual training data distribution?**
>
> We determine the data distribution by balancing language representation, model performance goals, and data quality/availability. In practice, this involves:
> 1. We ensure adequate coverage for each target language (Section 3.3) through controlled ablation, so the tokenizer learns frequent morphemes and avoids excessive fragmentation.
>
> 2. We shape the training data mixture to reflect target vocabulary proportions, allocating data to each script such that the learned vocabulary naturally converges toward the desired distribution. (Section 4.4)
>
>
> This approach ensures that the final tokenizer provides balanced cross-lingual representation while still optimizing for token efficiency and model performance in key languages.
>
> **Q4: What is the role of pre-tokenization in training a multilingual tokenizer?**
>
> Section 3.2 shows that replacing GPT-2 rules with LLaMA-4’s script-agnostic regex and applying Unicode-aware normalization reduces token fragmentation by 38 - 40% for Indic scripts (Table 1). To the best of our knowledge, the impact of pre-tokenization has not been systematically examined for Indian languages, despite their diverse scripts and rich morphology.
>
> Additionally, we show that unconstrained segmentation introduces sentence-crossing merges that destabilize generation, so we introduce sentence-boundary constraints, improving stability.
>
> These findings demonstrate that pretokenization is a first-class design choice, materially affecting token quality, stability, and multilingual robustness.
>
> **Q5: Does including multi-word expressions in tokenizer vocabulary help and what is the effective way to learn these multi-words?**
>
> In Section 5, we first compare a one-stage vs. two-stage curriculum and show that a dedicated second stage is clearly beneficial for learning multi-words as well as maintaining high fertility. Our Stage-2 process constructs multi-word units that capture frequent idioms and collocations. Through extensive ablations, we show that:
> - Stage-1 alone delivers SOTA fertility,
> - Stage-2 further improves multiword alignment and improves performance without harming semantic consistency.

---

> > ### Author Response · Authors · 2025-11-21
> >
> > While we adopt two-stage training as one component to include multi-word expressions, the core performance and efficiency gains do not arise solely from it. Our contributions extend significantly beyond the baseline method:
> >
> > **(a) Stage-1 tokenizer achieves SOTA, even without SuperBPE**
> >
> > As shown in Section 3.2 (Table 1), our Stage-1 tokenizer already achieves state-of-the-art intrinsic scores across most Indic languages. This establishes that the improvements do not rely solely on Stage-2 and demonstrates substantial contribution in the data design, pretokenization strategy, and allocation policy alone.
> >
> > **(b) A unified, generalizable tokenizer evaluation framework**
> >
> > We introduce a unified and fully reproducible evaluation framework that integrates four standard intrinsic metrics over a consistent cross-lingual benchmark dataset. This provides a standardized and reusable infrastructure for multilingual tokenizer evaluation, addressing a clear gap in current literature and enabling transparent, comparable experimentation for future research.
> >
> > **(c) Comprehensive ablations validating every design choice and extensive evaluation**
> >
> > Our paper includes end-to-end ablations on data volume, data distribution, pretokenization, vocabulary allocation, Stage-1 vs. Stage-2 contributions, and vocabulary quality (glitch analysis). This goes beyond applying an existing method, it provides explanatory scientific insight into what makes multilingual tokenization effective. In addition, we conduct extensive intrinsic and extrinsic evaluation, including four intrinsic metrics, full LLM pretraining using our tokenizer, downstream evaluation on nine English and seven Indic benchmarks, and end-to-end model throughput and latency analysis to quantify efficiency gains.
> >
> > **(d) Practical contribution: tokenizer replacement in pretrained models**
> >
> > Finally, we show that our tokenizer can directly replace the tokenizer of an existing pretrained LLM, preserving generation quality while significantly improving inference throughput (44% increase in throughput over base Llama-4 tokenizer).
> >
> > We therefore contend that our contribution is not limited in novelty. Our work offers a scientifically grounded and broadly applicable framework for multilingual tokenizer design, extendable well beyond Indic languages. More importantly, it highlights an underexplored but crucial aspect of LLM development, i.e. the design of multilingual tokenizers and we view this as a substantive step toward enabling more equitable and culturally inclusive multilingual LLMs.
> >
> > We have incorporated these clarifications directly into the introduction (Line 49) to make the updates clearly visible and to better highlight the novelty of our contributions.

---

> > > ### Author Response · Authors · 2025-11-27
> > >
> > > Dear Reviewer xsrk!
> > >
> > > Thank you once again for your thoughtful review. We believe the points outlined above directly address your question on technical novelty and clarify the broader scientific contributions of the work, and we would be glad to elaborate further if needed.
> > >
> > > Thanks!

---

### Official Review · Reviewer_4cy4 · 2025-11-02

**Soundness:** 3
**Presentation:** 2
**Contribution:** 2
**Rating:** 4
**Confidence:** 4

**Summary:**

The authors introduce an efficient tokenizer for Indic LLMs built on top of SuperBPE, achieving state-of-the-art parity scores across Indic languages, code, and English. Their proposed IndicSuperTokenizer combines the strengths of both subword and multi-word tokenization, along with script-specific pretokenization strategies that yield more linguistically aligned segmentations. When applied to train and evaluate models across English and multiple Indic languages, the tokenizer largely preserves model performance while boosting inference throughput by 44%

**Strengths:**

- Adapting the original SuperBPE algorithm to low-resource, non-Latin script languages like Indic is commendable and makes a meaningful contribution to the multilingual NLP community.
- The paper includes very fine-grained analysis and ablations, covering key factors in training superword tokenizers for Indic languages, such as vocabulary size, merging strategies, and pre-tokenization techniques.
- Using script-agnostic pretokenization in the first stage of tokenizer training improved token-to-word ratios by 38–40% on Indic scripts.
- Their tokenization approach preserves model performance while boosting inference throughput.

**Weaknesses:**

- Section 2.4 contains comparisons with different baseline toenizers. But is fertility fairly comparable here, given the potential difference in data distributions each tokenizer baseline was trained on? At the very least, a short description of how each tokenizer was trained, if possible, would make comparisons more meaningful.
- It would also really help to see a direct comparison between IndicSupertokenizer and a regular BPE trained on the same data with script-agnostic pretokenization. Right now, it’s hard to tell whether the improvements come from the tokenizer itself or just differences in the training data.
- For downstream evaluation, it’s not clear what results are zero-shot or from finetuning. I know that some of the Indic task-specific datasets have train/test splits.
- More on the downstream evaluation, is there a reason why IndicBPEtokenizer (just using the first stage) wasn’t considered for these extrinsic evaluations? I believe it would help to isolate the contribution of the super words learned in the later stage.
- In section 4.3, can you provide more details on the 200 examples used for analyzing inference efficiency? Are these parallel sentences? Also are these the same models evaluated in section 4.2?
- Compared to the original SuperBPE paper, the performance improvements here appear smaller. Any thoughts on why that might be would be valuable.

**Questions:**

- Do you have any insights from your analysis of vocabulary allocation strategies? Prior work has shown that training language-family or script-specific tokenizers and then merging their vocabularies can sometimes benefit low-resource languages. Though it increases vocabulary size, it can lead to small gains in downstream performance. You focus on fertility here, but it would be interesting to see whether similar trends hold for your setup.
- Why does English dominate in your dataset? Was that to preserve English performance in LLaMA or to encourage cross-lingual transfer to Indic languages, or was it just a result of data availability?
- Finally, pointing readers to the appendix for key metrics (as in line 42) isn’t ideal. It would be better to report an aggregated number, like average fertility, directly in the main text for readability.

---

> ### Author Response · Authors · 2025-11-21
>
> We thank the reviewer for the detailed and constructive feedback. We appreciate the positive recognition of our contributions, including the focus for designing multilingual tokenizer, comprehensive ablations, and the efficiency gains achieved.
>
> > **W1:** Fairness of fertility comparisons in Section 3.4
>
> While we agree that tokenizer training details and data distributions affect its fertility, there is limited information about training data distribution and training details. As suggested we have compiled all available documentation on tokenizers under study in Table 22 in Appendix C.5. We thank the reviewer for making our work stronger.
>
> | **Model**            | **Vocab Size** | **Training Algorithm / Framework**                  | **Data Distribution**                                                |
> |----------------------|---------------------|------------------------------------------------------|----------------------------------------------------------------------|
> | **Sutra**            | 256K           | BPE / SentencePiece                                  | Balanced multilingual dataset; uniform sampling to avoid English dominance |
> | **Sarvam-2B**        | 68K            | Not publicly disclosed; tokenizer unspecified        | Not publicly disclosed                                               |
> | **Gemma-3-27B-it**   | 262K           | SentencePiece                                        | Over 140 languages; balanced multilingual design                     |
> | **GPToss**           | 200K           | o200k_harmony (OpenAI BPE variant) / TikToken        | Not publicly disclosed                                               |
> | **LLaMA-4**          | 200K           | BPE                                                  | Not fully disclosed                                                  |
> | **Qwen3-32B**        | 151K           | Byte-level BPE                                       | Not publicly disclosed                                               |
> | **LLaMA-3.2-1B**     | 128K           | BPE / SentencePiece-based                            | Not publicly disclosed                                               |
> | **Mistral-Nemo**     | 131K           | Tekken tokenizer (TikToken-based)                    | 100+ languages; heavy code + multilingual text                       |
> | **DeepSeek-R1**      | 128K           | BPE (not published)                                  | Not publicly disclosed                                               |
>
> > **W2:** Direct comparison with a regular BPE trained on the same data
>
> We also evaluated our stage-1 tokenizer (trained with BPE on the same data and pretokenization). As shown in Section 3.2, Table 1, Row no. 2, our Stage-1 tokenizer, already achieves state-of-the-art (SOTA) performance across most Indic languages. Additional improvements are attributed to the tokenizer design choices arrived at after careful ablations.
> - Curation of high quality training data across 22 Indic languages (Section 3.3)
> - Refined pre-tokenization and normalization tailored to Indic linguistic structure (Section 3.2)
> - Overall vocabulary budget (Refer to ablations in Section 5)
> - Training data-mix across languages that leads to an efficient tokenizer for low-resource languages without sacrificing the tokenizer performance on English (Section 3.3, Section 4.4 - Table 10)
> - Corpus-driven (instead of explicit) vocabulary allocation for different languages (Section 4.4)
>
> We also provide the results here for easier reference:
> | **Tokenizer**                 | **as** | **bn** | **brx** | **code** | **doi** | **eng** | **gom** | **gu** | **hi** | **kas** | **kn** | **mai** | **ml** | **mni** | **mr** | **nep** | **or** | **pa** | **san** | **sat** | **snd** | **ta** | **te** | **urd** |
> |------------------------------|-------:|-------:|--------:|---------:|--------:|--------:|--------:|-------:|-------:|--------:|-------:|--------:|-------:|--------:|-------:|--------:|-------:|-------:|--------:|--------:|--------:|-------:|-------:|--------:|
> | **IndicSuperTokenizer (Stage-1)** | 1.83 | 1.74 | 1.99 | 1.54 | 1.56 | 1.33 | 2.17 | 1.83 | 1.36 | 1.36 | 2.15 | 1.56 | 2.24 | 2.27 | 1.61 | 1.59 | 1.65 | 1.47 | 2.51 | 3.60 | 1.45 | 2.07 | 1.83 | 1.47 |
> | **IndicSuperTokenizer (Stage-2)** | 1.85 | 1.74 | 2.04 | 1.47 | 1.45 | 1.12 | 2.17 | 1.77 | 1.23 | 1.21 | 2.19 | 1.58 | 2.30 | 2.28 | 1.63 | 1.62 | 1.65 | 1.39 | 2.59 | 3.72 | 1.45 | 2.12 | 1.88 | 1.44 |
>
> > **W3:** Downstream evaluation: shot settings / finetuning / splits
>
> All downstream evaluations were performed using fixed-shot prompting only, with no task-specific fine tuning. Concretely, we follow:
> - 25-shots: ARC-Challenge
> - 10-shots: HellaSwag
> - 5-shots: MMLU, GSM8K, TriviaQA
> - 5-shots: all Indic benchmarks
> - 0-shot: remaining English benchmarks
>
> We follow each dataset’s standard train/test split, which we have documented in detail in the Appendix B.2.

---

> > ### Author Response · Authors · 2025-11-21
> >
> > > **W4:** Why Stage-1 (IndicBPE) was not originally in extrinsic evals
> >
> > We appreciate the reviewer’s insightful suggestion. Stage-1 served as our controlled subword-based tokenizer baseline, and therefore we evaluated it primarily through intrinsic metrics. Subsequently, we have now trained a model with the Stage-1 tokenizer as well, and it also achieves strong downstream performance. We have provided these results in Table 25 in the revised manuscript.
> >
> > **Table: Comparison of downstream performance between IndicSuperTokenizer (Stage-1) and  IndicSuperTokenizer (Stage-2). Stage-1 uses BPE only.**
> >
> > ### English Benchmarks
> >
> > | **Dataset**        | **IndicSuperTokenizer (Stage-1)** | **IndicSuperTokenizer (Stage-2)** |
> > |--------------------|-----------------------------------|-------------|
> > | **Hellaswag**      | 0.348                             | 0.357       |
> > | **Commonsense QA** | 0.193                             | 0.204       |
> > | **OpenbookQA**     | 0.214                             | 0.218       |
> > | **Winogrande**     | 0.515                             | 0.510       |
> > | **GSM8K**          | 0.021                             | 0.018       |
> > | **ARC Easy**       | 0.625                             | 0.630       |
> > | **ARC Challenge**  | 0.279                             | 0.292       |
> > | **MMLU**           | 0.255                             | 0.249       |
> > | **DROP**           | 0.042                             | 0.036       |
> > | **Average**        | 0.277                             | 0.279       |
> >
> > ### Indic Benchmarks
> >
> > | **Dataset**              | **IndicSuperTokenizer (Stage-1)** | **IndicSuperTokenizer (Stage-2)** |
> > |--------------------------|-----------------------------------|-------------|
> > | **Indic COPA**           | 0.556                             | 0.556       |
> > | **Indic Sentiment**      | 0.557                             | 0.551       |
> > | **Indic XNLI**           | 0.366                             | 0.346       |
> > | **Indic Paraphrase**     | 0.562                             | 0.539       |
> > | **MILU**                 | 0.265                             | 0.258       |
> > | **ARC Challenge Indic**  | 0.247                             | 0.244       |
> > | **Trivia QA (Indic)**    | 0.268                             | 0.262       |
> > | **Average**              | 0.403                             | 0.394       |
> >
> >
> > These results confirm that the Stage-1 tokenizer itself offers a strong baseline. With stage-2 training, we see a comparable performance with around 3% gain in the inference throughput (due to improved fertility).
> >
> >
> > > **W5:** Details on 200 examples and models across extrinsic & latency Experiments
> >
> > Yes, the same model checkpoint was used for both:
> > 1. All extrinsic downstream evaluations (Section 4.2), and
> > 2. All latency/throughput measurements (Section 4.3).
> >
> > No intermediate finetuning, hyperparameter changes, or sampling-strategy modifications were introduced, ensuring strict comparison between accuracy and efficiency results.
> > Re the 200 examples used in Section 4.3:
> > - They are not parallel sentences, but a balanced multilingual evaluation set designed to mimic realistic inference workloads.
> > - Specifically, the set contains 20 sentence-completion inputs per language, across English + 9 major Indic languages.
> >
> > Below we provide input sequence length distribution of those 200 examples:
> >
> > | **Model**              | **min** | **p75** | **p90** | **p99** | **max** | **avg** |
> > |------------------------|--------:|--------:|--------:|--------:|--------:|--------:|
> > | **Llama-4**            |   288   |   805   |  1157   |  2583   |  2869   |   784   |
> > | **IndicSuperTokenizer**|   178   |   440   |   541   |   654   |   676   |   379   |
> >
> > We have clarified these details in Appendix Section C.4.
> >
> > > **W6:** Smaller gains compared to original SuperBPE paper
> >
> >
> > We want to clarify that the smaller gains relative to the original SuperBPE paper primarily arise from **differences in scale** as well as the added complexity of including **multiple low-resource languages**, rather than limitations of our approach. Extending tokenizer improvements across multiple scripts and morphologically diverse languages is non-trivial, and the “curse of multilinguality” [1][2] is well documented. The original SuperBPE evaluation used a 7B model trained on ~300B predominantly English tokens, whereas our setting uses a 1B model trained on ~50B multilingual tokens (280 H100 hours) across English and 10 Indic languages, majorly constrained by compute budgets.
> >
> > Despite these factors, we observe consistent gains in intrinsic metrics including fertility, NSL, as well as latency and throughput, indicating that our improvements are robust and are likely to scale further with larger models and corpora.
> >
> >
> > [1] Gurgurov et al Multilingual large language models and curse of multilinguality CoRR abs/2406.10602
> >
> > [2] Chang et al "When is multilinguality a curse? language modeling for 250 high-and low-resource languages." EMNLP 2024

---

> ### Author Response · Authors · 2025-11-21
>
> **Q1: Insights on vocabulary allocation strategies / merging tokenizers**
>
> We examined both strategies (1) training script-specific tokenizers and explicit merging;  and (2) corpus-driven alignment, where we jointly train a single tokenizer on the concatenated multilingual corpus (Refer to Section 4.4).
>
> Our results show that vocabulary allocation via corpus-driven alignment yields lower fertility and stronger intrinsic performance for low-resource languages compared to explicit merging. Joint training implicitly allocates vocabulary across scripts, producing a more balanced distribution of shared vs. language-specific tokens while also offering a more scalable multilingual tokenization strategy.
>
> Tokenizer merging, on the other hand, requires explicit script-wise vocab budgets, training of multiple tokenizers, and rule-based heuristics to reconcile vocabularies. Our experiments show that such explicit merging is less effective and often inefficient.
>
>
> **Q2: Why English is dominant in our dataset**
>
> Our design intentionally includes a higher proportion of English data to preserve transferability and maintain strong performance on high-resource languages, which is an established requirement for multilingual LLMs. At the same time, the data mixture ensures that all 22 Indic languages receive sufficient representation to support generalization across scripts and linguistic families.
>
>
> **Q3: Reporting aggregated numbers in main text**
>
> Thank you for the suggestion. The revised manuscript now includes reference to main text Section 3.5 Table 3.
>
>
> We thank the reviewer again for the constructive feedback which helped us improve clarity, include important analysis, and strengthen the experimental record. We hope these clarifications and the new results address your concerns.

---

> > ### Author Response · Authors · 2025-11-27
> >
> > Dear Reviewer 4cy4!
> >
> > Thank you for your valuable insights and questions. Your feedback helped strengthen our work particularly by motivating clearer explanations of both our benchmark design and our tokenizer training strategy.
> >
> > We hope the updates comprehensively address all of your comments.
> >
> > Thanks!

---

### Author Response · Authors · 2025-12-03
**Final General Comment**

We sincerely thank the reviewers for their careful evaluation and for highlighting several strengths of our work, including:

- Importance and timeliness of the task: the inefficiency of standard tokenizers for morphologically rich, non-English languages.
- Strong intrinsic performance gains, with IST achieving new SOTA fertility, improving **39.5% over LLaMA-4** and **18% over Sutra** (the current SOTA in Indic tokenizers) on average.
- Demonstration that pre-training a 1B model from scratch with IST yields a **44% improvement in inference throughput** including gains across downstream task performance.
- Extensive ablation studies validating each design component including vocabulary size, data scaling, merging strategies, and pre-tokenization alongside thorough latency, throughput, and efficiency comparisons.
- Evidence that IST can be substituted into existing pre-trained models, preserving their performance while substantially improving throughput efficiency.
- Comprehensive and well-structured presentation, including analyses of “glitch tokens”.

**Broader Impact**

Most existing tokenizer research remains centered on English and Western languages, leaving low-resource and morphologically diverse languages insufficiently represented. Our work directly addresses this important gap by introducing a unified multilingual tokenizer design that improves linguistic coverage, reduces fragmentation, and yields more consistent subword representations across diverse scripts. By enabling more efficient downstream training and ensuring fairer representation of minority languages, our tokenizer helps reduce performance disparities in multilingual LLMs.

Beyond the tokenizer itself, our work offers a **systematic, generalizable recipe** for building efficient multilingual tokenizers. We provide concrete answers to key design questions on data distribution, pre-tokenization choices, joint training vs. merging tokenizers, and the incorporation of multi-word expressions. These findings generalize well beyond Indic languages and offer practical guidelines for multilingual and multi-script tokenization. Our novel contributions include:

- A unified, practical and reproducible multilingual tokenizer evaluation framework and benchmark
- A curriculum-based multi-word learning strategy tailored for multilingual tokenizers.
- Practical methodology for tokenizer replacement in existing pretrained models.
- Comprehensive ablations validating every design choice, supported by extensive intrinsic and downstream evaluation.


Our study explicitly investigates several fundamental research questions:

- How can we improve low-resource language performance without hurting high-resource performance?
- Should we train language specific tokenizers and merge them or should we train a  joint tokenizer?
- How do we decide the multilingual training data distribution?
- What is the role of pre-tokenization in training a multilingual tokenizer?
- Does including multi-word expressions in tokenizer vocabulary help and what is the effective way to learn these multi-words?

---

**Rebuttal Summary**

During the rebuttal period, we undertook substantial effort to address the reviewers’ concerns through additional experiments, clarifications, and manuscript improvements.

***Key Experiments***

1. Stage-1-only tokenizer downstream evaluation: We trained a model using only the Stage-1 BPE tokenizer (same data) which provides a strong baseline while Stage-2 yielding improved inference throughput.

2. Morphology-aware tokenizer latency analysis: We evaluated end-to-end latency for the morphology-aware tokenizer. Our measurements confirm that the morphology-aware tokenizer has 53.28% lower throughput, validating the high latency we previously noted. Naive morphology-aware pipeline is thus impractical without a more efficient implementation, which could be interesting for future work.


***Manuscript Revisions***

- Expanded Introduction with explicitly stated research questions.
- Detailed descriptions of all baseline tokenizers.
- Inclusion of Stage-1 downstream results for a direct comparison.
- Added morphology-aware tokenizer latency experiment results.
- Added full details on the 200-example latency benchmarking dataset.

We appreciate the reviewers’ constructive feedback. In summary, our work contributes both scientific insights and a reproducible, practical framework for multilingual tokenizer development, addressing a crucial yet understudied component of modern LLM pipelines. The additional experiments and clarifications provided during rebuttal substantially strengthen the manuscript and underscore the significance and generality of our contributions.

---

### Note · Authors · 2026-01-06

**Comment:**

Withdrawing this work to avoid the breach of dual submission policies of the conferences.

**Withdrawal Confirmation:**

I have read and agree with the venue's withdrawal policy on behalf of myself and my co-authors.